

# Spatial characteristics of frazil streaks in the Terra Nova Bay Polynya from high-resolution visible satellite imagery

Katarzyna Bradtke[1], Agnieszka Herman[2]

[1]Faculty of Oceanography and Geography, University of Gdańsk, Gdańsk, Poland
[2]Institute of Oceanology, Polish Academy of Sciences, Sopot, Poland

*Correspondence to*: Katarzyna Bradtke (katarzyna.bradtke@ug.edu.pl)

**Abstract.** Coastal polynyas around the Antarctic continent are regions of very strong ocean–atmosphere heat and moisture exchange, important for local and regional weather, sea ice production and water mass formation. Due to extreme atmospheric conditions (very strong offshore winds, low air temperature and humidity) the surface ocean layer in polynyas is

highly turbulent, with mixing due to combined Langmuir, wind-induced and buoyancy-driven turbulence. One of the visible signs of complex interactions between the mixed layer dynamics and the forming sea ice are frazil streaks, elongated patches of high ice concentration separated by areas of open water. In spite of their ubiquity, observational and modelling analyses of frazil streaks have been very limited, largely due to the fact that their significance for heat flux and ice production is only just becoming apparent. In this study, the first comprehensive analysis of the spatial variability of surface frazil

concentration is performed for the Terra Nova Bay Polynya (TNBP). Frazil streaks are identified in high-resolution (pixel size 10–15 m) visible satellite imagery, and their properties (surface area, width, spacing and orientation) are linked to the meteorological forcing (wind speed and air temperature). This provides a simple statistical tool for estimating the extent and ice coverage of the region of high ice production under given meteorological conditions. It is also shown that the orientation of narrow streaks tends to agree with the wind direction, suggesting the dominating role of the local wind forcing in their

formation. Very wide streaks, in turn, deviate from that pattern, as they are presumably influenced by several additional factors, including local water circulation and the associated convergence zones. An analysis of peak wave lengths and directions determined from the images, compared to analogous open-water wave lengths computed with a spectral wave model, demonstrates a significant slow-down in the observed wave growth in TNBP. This suggests an important role of frazil streaks in modifying wind-wave growth and/or dissipation in polynyas.

**1 Introduction**

Coastal (or latent heat) polynyas are important elements of the Southern Ocean sea ice landscape. They are created mechanically by wind (possibly combined with currents) leading to sea ice divergence at the coast and its export in the offshore direction. Polynya's development is often additionally supported by limited advection of extraneous ice pack blocked by islands, peninsulas, ice tongues, grounded icebergs, or other (semi)permanent coastal features (Morales Maqueda



et al., 2004). Due to very strong ocean–atmosphere heat and moisture fluxes and high rates of new ice production with the associated brine rejection, latent heat polynyas play an important role in shaping the local and regional weather patterns, as well as in water mass formation, ocean mixing and baroclinic processes. Understanding polynya evolution, dynamics and thermodynamics, and the ocean–sea ice–atmosphere interactions involved, is crucial for developing parametrizations of the relevant processes, and thus for improving the performance of weather, sea ice, ocean and climate models of the polar

regions. Of particular interest is estimation of, first, the net ocean–atmosphere heat flux, and second, the sea ice production rates and the associated total ice volume formed within a polynya – quantities that are very sensitive to the properties of the sea surface (in particular, its roughness), as well as to the state of the atmospheric boundary layer (ABL) and the ocean mixed layer (OML). These properties, in turn, depend on the amount and spatial distribution of frazil and grease ice within the surface layer of the polynya, as well as its surface area. Thus, a feedback mechanism is formed, with sea ice production

and distribution both affected by and affecting the dynamics and heat transfer in the ABL–sea ice–OML system. On one branch of that feedback loop, formation and growth of ice crystals and their agglomerates, as well as their transport within the OML, are governed by OML's temperature structure and dynamics: Ekman currents, Stokes drift, buoyancy-induced mixing and Langmuir turbulence (e.g., Belcher et al., 2012; Herman et al., 2020; in the context of OML material transport in general, see Chamecki et al., 2019). As a result, complex, three-dimensional patterns emerge, with ice crystals reaching

depths of meters or even tens of meters (Drucker et al., 2003, Ito et al., 2019, 2020, 2021, Ohshima et al., 2022). At the surface, they tend to accumulate in convergence zones between neighbouring Langmuir cells, forming characteristic elongated streaks of grease ice separating areas of low crystal concentration. Closing the feedback loop, the strong local gradients of ice concentration produce gradients of bulk density and viscosity of the ice–water mixture, leading to suppressed turbulence and reduced sea surface roughness within streaks. The fact that high concentrations of frazil attenuate short waves

is well known (hence the term 'grease ice'). Recent measurements show that a substantial part of the total ocean–atmosphere heat flux over polynyas takes place through spray within the lowest ABL (as opposed to flux directly through the sea surface; Guest, 2021a,b), so that suppression of wave breaking and spray generation by frazil streaks has the potential of drastically reducing the heat exchange (Ackley et al., 2022). Hence, the knowledge of the fraction of sea surface covered by streaks in function of the atmospheric forcing and distance from the shore is crucial for reliable estimation of that exchange.

Similarly, it is reasonable to assume that by eliminating short waves from the wave energy spectrum, streaks modify the growth and evolution of wind waves in polynyas.

In spite of recent progress in developing parameterizations of new ice formation (e.g., Wilchinsky et al., 2015, Yue and Shen, 2021), many of the processes and interactions mentioned above are rather poorly understood at present and therefore their effects cannot be taken into account in models. The remoteness of the affected regions and harsh conditions during

polynya events make conducting *in situ* measurements there a big challenge. This makes satellite observations particularly valuable. Indeed, most available information on polynyas' occurrence, variability, and ice production have been obtained from satellite data. The most widely used technique is a passive microwave (PMW) radiometry, due to its independence from sunlight and cloud cover as well as frequent coverage and continuity of observations since the 1970s (Hollands and



Dierking, 2016, and also references therein, Nakata et al., 2021). Nevertheless, coarse spatial resolution of PMWs sensors
(2.5–25 km, dependent on instrument and product) does not allow to accurately determine the size and shape of the polynya.
Several studies have shown the usefulness of satellite thermal imaging (TIR) to characterize area of the polynya more
accurately (Ciappa et al., 2012, Preußer et al., 2015, Hollands and Dierking, 2016, Aulicino et al., 2018, Vincent, 2019).
Most TIR radiometers operating from near polar orbits (e.g. MODIS, AVHRR, SLSTR) have a swath width of about 1.5–3
thousands km and a resolution of approximately 1 km at nadir to a few km at the swath edges. The drawback of this satellite
technique is the limitation of useful information to cloudless areas. Moreover, to acquire more detailed spatial information
about the ice distribution and condition even higher spatial resolution is needed. Hollands and Dierking (2016) emphasize
the potential of synergy of satellite data from different techniques, including synthetic aperture radar (SAR) and optical
radiometers (VIS), as a complementary data source, for increasing the possibilities of characterizing ice in polynyas. Both
techniques can deliver data with a resolution of 10–150 m, but at the cost of a smaller swath width compared to the lower
resolution techniques mentioned above (moreover, VIS sensors can be used only at daylight in cloudless conditions). Hence,
they are more useful for describing the characteristic properties of the small scale features like frazil streaks (e.g. Ciappa and
Pietranera, 2013) or pancake ice (e.g. Aulicino et al., 2019), than for continuous monitoring of polynya dynamics.

Arguably, the presence of frazil streaks at the sea surface is an indicator of a certain 'regime' of the OML (extreme
atmospheric forcing, with very high wind speeds and very low air temperatures; Langmuir turbulence associated with short,
fetch-limited waves; frazil ice formation; etc.). Therefore, frazil streaks detected in satellite imagery can be used as a proxy
of those processes – provided a relationship is known between streaks properties and the state of the OML. Obviously, a
necessary prerequisite for developing such parametrizations is development of useful quantitative measures characterizing
the streaks.

Frazil streaks have been observed *in situ* and in satellite imagery in polar regions of both hemispheres, e.g. in the Weddell
Sea (Eicken and Lange, 1989), Terra Nova Bay (Ciappa and Pietranera, 2013, Hollands and Dierking, 2016, Thompson et
al., 2020), or Chukchi Sea (Ito et al., 2019). They have also been studied in a laboratory (Dethleff et al., 2009) and simulated
numerically (Matsumura and Ohshima, 2015, Herman et al., 2020). In a wider context, bands of positively buoyant particles
(so called windrows) accumulating within surface convergence zones associated with Langmuir turbulence are common at
all latitudes. However, the accounts listed above (for a more complete overview, see Herman et al., 2020) are limited to
qualitative reports of the presence of frazil streaks or their most basic characteristics, like e.g. the average/typical distance
between neighbouring bands, or probability distribution of distances (which, according to observational and theoretical
studies, often is log-normal, see, e.g. Qiao et al., 2009, Csanady, 1994). Apart from the few numerical studies cited above,
modelling of frazil formation has been limited to one-dimensional simulations in the water column, i.e., disregarding any
variability in horizontal plane (e.g., Omstedt and Svensson, 1984, Holland and Feltham, 2005, Heorton et al., 2017, Rees
Jones and Wells, 2018). Coupled sea ice–hydrodynamic models of polynyas (e.g., Wang et al., 2021) have insufficient
spatial resolution to capture frazil–OML interactions at the scale of individual streaks. To the best of our knowledge, no
quantitative, systematic analysis of frazil streaks has yet been published.



Previous works based on satellite data, in which frazil streaks were identified, utilised SAR images (Ciappa and Pietranera, 2013, Nakata et al. 2015, Hollands and Dierking, 2016). Operating at microwave frequencies, SAR systems provide data

irrespective of cloud cover and daylight, in contrast to optical sensors. The SAR signal is sensitive to the roughness of the surface, allowing the separation of open polynya from the consolidated ice, as well as open water from the frazil streaks inside the polynya, or classification of different ice zones outside. However, single examples of SAR and high resolution optical data comparisons show that the contrast between water and frazil ice in the visible images is much greater (Hollands and Dierking, 2016, Herman et al., 2020), which allows for a more accurate characterization of geometric features of the

streaks. This is true especially for streaks with a width close to the data resolution, the distinction of which in SAR images is hampered by the speckle effect resulting from coherent wave interference. The lack or low intensity of sunlight and cloud cover significantly limit the availability of satellite data in the visible spectrum for the polar regions. However, in the last decade, the EO-1, Landsat-8 and Sentinel-2 satellites equipped with high-resolution optical radiometers recorded many scenes in the polar regions, which enable first statistical characterization of spatial properties of frazil streaks. Two

constellations of pairs of satellites Sentinel-2 A and B and Landsat-8 and -9 currently operating from near polar orbits will allow to extend these possibilities in the future.

In this study, an analysis of frazil streaks in 32 polynya events in the Terra Nova Bay (TNB; Fig. 1) is performed based on high-resolution (pixel size 10–15 m) visible satellite imagery. For each image, polynya size, ice concentration, and geometric properties of streaks are determined and related to the observed air temperature $T$ and wind speed $U_w$ from an

automatic weather station. Strong correlation between the atmospheric forcing and all analysed variables is found. In particular, the polynya extent and surface area, as well as changes of ice concentration with offshore distance $X$ can be estimated from $T$ and $U_w$ with simple empirical formulae. As expected, ice concentration increases with $X$ faster at lower $T$. Less obviously, the increase of frazil concentration with distance is negatively correlated with $\underline{U}_w$ as well, suggesting that, with increasing wind strength, the effects of faster advection dominate over effects of higher wind-induced ocean heat loss.

Additionally, for a subset of scenes in which wind waves are discernible, peak wave length and direction are determined and compared with corresponding open-water wave growth curves computed with a spectral wave model. It is shown that the increase of wavelength with fetch in polynyas is substantially slowed down, and that only a small fraction of this slowdown can be explained by a decrease of wind speed with distance from shore. Thus, wave interactions with frazil/grease ice are the most plausible explanation for the observed patterns of wave growth. Finally, an analysis of streaks' orientation reveals a

characteristic, repeatable fan-like pattern. The dominant role of the wind in shaping the streaks decreases with the accumulation of frazil ice and for the bands with an average width of more than 1 km other factors become more important.



## 2 Data and methods

### 2.1 Dataset

The study is based on data from three satellite sensors: ALI (Advanced Land Imager), OLI (Operational Land Imager) and
MSI (Multispectral Instrument), operating respectively on the EO-1, Landsat-8 and Sentinel-2 (A and B) satellites. Standard
Level 1 products (radiometrically and geo-corrected radiance at the top of the atmosphere) were obtained via the following
services: USGS Earth Explorer (ALI and OLI) and Copernicus Open Access Hub (MSI). All three sensors record radiation
reaching the satellite in several spectral bands in the visible (VIS), near infrared (NIR) and short-wave infrared (SWIR)
range. ALI and OLI are equipped with an additional panchromatic channel (PAN) which, due to the highest spatial
resolution, was used to distinguish the frazil streaks. In the case of the MSI sensor, data from RGB channels were averaged
as an equivalent. The surface covered with frazil streaks was extracted by image classification, taking into account the
contrast between water and ice caused by different light reflectance. Additionally, SWIR data was used to prepare the cloud
mask and assess the possible impact of the atmosphere on the classification result, as described further in Section 2.2. The
characteristics of the satellite products and the spectral channels used in the analysis are presented in Tab. 1. For the TNBP
region, 32 scenes with visible frazil streaks were obtained, recorded in the years 2009–2021, mainly in September and
October (Tab. 2; Suppl. Fig. S1). To investigate the relationship between the properties of the frazil streaks and the
conditions in which the polynyas are formed, meteorological data from the automatic weather station Manuela
(https://amrc.ssec.wisc.edu/aws/), located on the Inexpressible Island (see Fig. 1), were used. Unfortunately, data on wind
speed and direction were not available for the scenes from the year 2009. Both the instantaneous data recorded just before
the satellite's overpass, and the statistics of 10-minute records from longer periods were analysed. The best correlation with
frazil properties was obtained for the characteristics averaged over the 24 hours prior to the satellite registration, and only
these results will be presented further. Only the wind direction is taken from the last measurement prior to the satellite
overpass. In the considered situations, the temperatures range from −10 °C to −30 °C, and the velocities of winds blowing
from SW to WNW (250°–307°) in the range of 10–34 m·s$^{-1}$ (Tab. 2). Speeds above 17 m·s$^{-1}$, characteristic of catabatic
winds, dominate and are accompanied by temperatures below −20 °C.

### 2.2 Methods

#### 2.2.1 Satellite data processing

The spectral data was pre-processed in a standard way by converting the original values to dimensionless reflectance, and
then resampled to a 10 m raster grid in the UTM zone 58S coordinate system (EPSG:32758) if it was in another. The contour
of the polynya was determined by manually digitizing the image. The downwind edge was marked by a rim separating the
frazil streaks from the consolidated ice, which in most cases was clearly visible for the part of polynya (Fig. 1b). Where the
boundary was blurred (Fig. 1c), the determinant was the presence of elongated open water 'ponds' trapped between freezing



streaks. The boundary of the ice sheet was drawn in such a way as to include as well the areas of landfast ice along the shore.
Comparing the results of the OLI sensor with data from the Thermal Infrared Sensor (TIRS), the second instrument on board

the Landsat 8 satellite, shows a good agreement of the boundary drawn in this way with the zone of rapid temperature change between the warm open polynya and the colder consolidated ice. It is only the presence of wide zones of frazil accumulation/transformation, where the streaks are no longer visible, that may cause difference in the determination of the area of the polynya on the base of optical and thermal data, which in the one worst case reaches 18%. In other cases, the differences did not exceed a few percent. Most of the scenes were cloudless, or despite the presence of clouds, it was

possible to determine polynyas' boundaries. In a few cases, however, the recorded scene does not cover the entire polynya (Tab. 2).

In the next step, a cloud mask was created by segmentation of the SWIR image, using spectral band about 1.6 μm, present in all three sensors. In this spectral range, the reflectance of sea ice is close to zero, as is the reflectance of water, while for clouds and haze it remains high. To perform segmentation the eCognition software for image interpretation with the Multi-

Resolution Segmentation (MRS) algorithm was used (Definiens AG, 2012). This iterative algorithm merges consecutively neighbouring objects (starting from individual pixels) taking into account the similarity between adjacent objects (homogeneity criterion) (Definiens AG, 2012, Mesner and Ostir, 2014). The process stops when the resulting objects achieve the maximum allowed heterogeneity, expressed by scale parameter. To distinguish clouded areas medium scale segmentation (scale parameter 500) with a homogeneity criterion based solely on brightness in the SWIR channel were used. Segments

with an average reflectance above 0.1 were masked as covered by clouds. In a few cases, additionally, a relatively smaller increase in pixel brightness was observed in the SWIR channel associated with the presence of haze or thin clouds, which did not affect the ability to distinguish the streaks and determine their shape, but reduced the contrast between them and open water, which could affect results of image classification. For this reason, an additional level of coarse segmentation (scale 1000, average SWIR reflectance and its variability over segment as homogeneity criteria) was created for the cloud-free area.

These segments were manually classified, by visual inspection of SWIR and PAN images, into up to three regions: (i) cloud shadow (reflectance reduction in both spectral bands), (ii) haze or thin clouds (increase in SWIR reflectance and its variability over segment, reduced contrast in PAN band) and (iii) 'clear', i.e. free of atmospheric influences.

In order to classify pixels inside regions of the polynya into open water and ice categories, Contrast Split Segmentation algorithm (Definiens AG, 2012, Mesner and Ostir, 2014) was used. This algorithm evaluates the optimal threshold for

dividing the scene into dark and bright objects that maximizes the contrast between them. Since the analysed images were not subjected to atmospheric correction, the edge contrast ratio mode was used to calculate the contrast. This mode uses the difference between reflectance of possible bright and dark objects normalized to the sum. The classification was performed separately for each region in two stages. In the first stage, bright objects were classified as frazil streaks. The remaining pixels were classified again in a second step where dark objects were assigned to class 'open water'. The remaining pixels of

medium brightness were left to be decided after visual inspection of the results. Mostly, these were pixels at the edges of wider bands or pixels forming narrow, less contrasted streaks, so they were also classified as frazil. However, in a few



images, this approach would result in merging the visible streaks into large patches. Therefore, in these cases, the intermediate pixels were included into the category of open water (cases marked in Tab. 2 as classified using adjusted scheme). Finally, pixels belonging to the same class in different regions (cloud shadow, haze, 'clear') were merged. Despite

the separate classification, the frazil streaks maintain continuity between regions, which is the result of the relatively high contrast between water and ice, regardless of the overall brightness of the image parts.

### 2.2.2 Data analysis methods

Based on the contour delineating boundary of the polynya, the area ($S_p$), maximum extent ($L_e$) as an Euclidian distance from the shoreline to the most distant point of the polynya, and length of border along the ice sheet ($L_b$) have been calculated (Tab.

2). In order to characterize frazil concentration and its spatial variability three metrices has been determined taking into account area of the open water and the frazil streaks in the cloudless part of the polynya: (i) polynya-averaged concentration ($C$); (ii) zonal-averaged concentration ($C_Z(X)$) calculated for 500 m wide zones parallel to the shore, whose distance from the shore ($X$) is determined in the middle of its width; and (iii) spatially-averaged concentration $C_S(x,y)$ calculated within moving windows of size 256×256 pixels (2.5×2.5 km). The distance from the shore to the nearest frazil streak ($X_{FS}$) was measured

along the shoreline every 20 m perpendicularly to its general direction. The regression of these parameters with the wind speed and air temperature was studied for the purpose of the parameterization of the polynya size characteristics and frazil concentration. Due to the mutual correlation of considered meteorological parameters, in the first step, multiple regression was used and partial correlations were examined. Further, simple linear relationships were determined with the selected parameter, which better explains the variability of the characteristics of the polynya. The root-mean-square error (RMSE)

and the relative root-mean-square error (RRMSE) were calculated to evaluate the results of parameterization. To characterise geometric features of frazil streaks, their orientation ($\theta_{FS}$), width ($W_{FS}$) and the distance between adjacent streaks has been calculated. In order to characterise orientation of the streaks and their spacing, their contours were smoothed and the skeletal lines were extracted as main lines of polygons using ESRI ArcGISPro software. Frazil streaks with an area of less than 1000 m$^2$ (10 pixels) or adjacent to clouds were omitted from the analysis. Due to the change in the orientation of the streaks, their

widths and distances between them were determined along the Y axis of the raster grid (UTM projection) with 10 m distance between cross sections in the X direction. Typical values of streaks' properties were characterized by non-parametric statistics, i.e. median and interquartile range. Probability density distributions of the frazil streaks' widths and spacing were calculated for logarithmically transformed values, due to the strong skewness of the distribution. Orientation was calculated for each segment of the streak skeleton as the angle, measured clockwise, between the North and the segment oriented

towards downwind edge of polynya. Additional attributes of the segment, such as its length and frazil streak's width (geometric mean of the streak cross sections along the segment) were used in the statistical characteristics of the orientation variability. To show spatial trends of the frazil streaks orientation, skeleton lines were rasterized and median values in the moving windows of 512×512 pixels (5.1×5.1 km) were computed. The lack of another source of observations based on the optical properties of water and ice at a similar or higher spatial resolution makes it impossible to validate the applied method



of frazil streaks extraction. For the three events, there are coinciding pairs of MSI/OLI images, recorded at a time interval of no more than 0.5 hours. Comparing determined area of frazil and open water shows differences of 5-6% in the assessment of frazil concentration. Despite a small difference in the recording time, the location of the streaks and their geometry changes, which is clearly visible in the analysed images and makes it difficult to evaluate the other results.

In order to extract wind-wave information from the panchromatic images, a two-dimensional (2D) fast Fourier transform (FFT) has been applied to each scene. Prior to the analysis, the image was locally standardised (across the cloud-free area) using moving window filter of 5×5 pixels to reduce the contrast between open water and frazil ice and to enhance small-scale variation. Spectra were computed within moving windows of size 256×256 pixels and smoothed with a moving-average filter of length 9. For each spectrum, its peak frequency and the corresponding peak wavelength, $L_p$, was determined, as well as the mean wave direction at the peak wavelength, $\theta_p$. Contrary to the remaining characteristics of the spectra, which were very sensitive to the details of the algorithm (size of the FFT window, location of that window relative to frazil/open water patches, etc.), the values of $L_p$ and $\theta_p$ remained stable within a wide range of parameters, indicating that they could be determined in a robust way (with an exception of the first few kilometers from the coast, where the half-wavelengths are comparable with the pixel size; see Suppl. Fig. S8 and, especially, S9; as the boundary of the regions with spurious $L_p$ are clearly seen in the maps in Fig. S9, they were used to mask those regions before further analysis, see Fig. 310a,b for examples). Moreover, very small differences in the results from the three coinciding MSI/OLI image pairs available show that, in spite of resolution differences, reliable values of $L_p$ and $\theta_p$ can be obtained from both data sources. Therefore, in terms of wind waves our analysis is limited to $L_p$ and $\theta_p$. Crucially, no information on wave height can be obtained from the type of imagery used here, as no clear relationship exists between pixel values and sea surface elevation or slope.

Among all images available, in 18 cases (marked in Suppl. Figs.S8 and S9) the size and geometry of the polynya allows computation of wave growth curves $L_p(X)$, where $X$ denotes distance from the coast measured in the wave direction along the (approximate) central axis of the polynya. For each of those cases, data from a 2-km wide strip was used, averaged over the strip width.

A one-dimensional (1D) version of the spectral wave model SWAN, version 41.31 ([https://swanmodel.sourceforge.io/](https://swanmodel.sourceforge.io/)) was used to obtain open-water peak wavelengths $L_{p,OW}(X)$ corresponding to those estimated from the satellite imagery. To this end, several deep-water, fetch-limited wave growth curves were computed with SWAN for a range of wind speeds between 10 and 40 m·s$^{-1}$ and with three different wind input/whitecapping source term combinations available in SWAN (according to the models of Komen et al. 1984, Jansen 1991, and Rogers et al., 2012, respectively). Based on those results, a mean open-water growth curve was determined by least-square-fitting a function $L_{p,n}=aX_n^{b}$, where $L_{p,n}$ is the dimensionless wavelength, $L_{p,n}=L_p g/U_w^2$, and $X_n$ is dimensionless wind fetch, $X_n=Xg/U_w^2$ ($g$ denotes acceleration due to gravity; see, e.g., Holthuijsen, 2007).



## 3 Results

### 3.1. Polynya size and polynya-averaged frazil concentration

The analysed polynyas are characterized by various shapes and sizes (Suppl. Fig. S1). They cover an area from 160 km$^2$ to over 1,900 km$^2$, and the downwind edge of the polynya at the most distant point from the shore reaches 13−65 km (Tab. 2,

Fig. 2). Both quantities, area ($S_p$) and maximum extent ($L_e$), strongly correlate with each other ($r$=0.87, $p$-level=0.0000). Polynyas with the largest area, exceeding 1000 km$^2$, which extend up to 30−60 km, are observed when the average wind speed exceeds 24 m·s$^{-1}$. With similar wind conditions, however, consolidated sea ice pack may locally block the development of the polynya, the shape of which becomes less regular and the area relatively smaller in relation to the maximum extent. In the analysed situations, the ratio of $S_p$ to $L_e$ vary from 8.7 to 43.4, and the length of polynya along the

ice sheet from 25 to 80 km. Consolidated sea ice extending far towards the ice sheet frequently splits polynyas into 2−3 parts. The surfaces classified as covered with frazil streaks are shown in Suppl. Fig. S2. Their area range from 35% to 75% of the analysed surface of the polynya, depending on the meteorological conditions (Fig. 2). The lowest polynya-averaged ice concentration ($C$), below 50% of the area, is accompanied by relatively low wind speed (lower than 17 m·s$^{-1}$) and high temperature (above −20 °C). The largest concentration, above 70%, is observed when the average wind speed exceeds 30

m·s$^{-1}$.

Table 3 summarizes results of the regression analysis which relates the parameters characterizing size of the polynya or frazil concentration to the parameters describing the meteorological conditions in which polynyas were formed. The observed variability of the area, maximum extent and polynya-averaged frazil concentration is consistent with the variability of the average wind speed from the 24 hours preceding satellite overpass. However, the area of polynya is less determined by

meteorological conditions than its maximum extent. The extent and frazil concentration are also significantly correlated with the average temperature (positively and negatively, respectively). Nevertheless, the relationship with the temperature was weaker than with wind speed and it seems that neglecting the influence of the former only slightly worsens the estimation of these parameters, RRMSE increases by about 1% and 2.3% for $C$ and $L_e$, respectively (compare multiple and simple regression statistics in Tab. 3, Eq. 1−6).

### 3.2. Spatial variability of frazil concentration


The area covered with frazil ice changes with the distance from the ice sheet. Within the first kilometer, open water generally accounts for more than 85% of the area. Streaks begin to be visible 0.5−1.7 km from the ice sheet (Fig. 3a). The stronger the wind and the lower the temperature, the shorter the distance from the coast (Tab. 3, Eq. 7−8). However, variability in the width of open-water zone along the ice sheet border should be noted. When the temperature is relatively high, higher than

−20°C, and winds with speeds below or close to 17 m·s$^{-1}$ (e.g. 26, 29 Oct 2016, 26 Feb 2018, 11, 12 Nov 2020, 8 Oct 2021) the distance to the frazil streaks from the ice sheet is larger and more varied in different parts of the polynya, often exceeding



2 km (Suppl. Fig. S2). Such situations occur especially in areas where the growth of the polynya is blocked (near the Drygalski Ice Tongue or in the northern part of the TNB). In these regions, the spatially-averaged (in 2.5×2.5 km moving window) frazil concentration $C_S(x,y)$ below 0.1 is maintained often to the border of the polynya (Suppl. Fig. S3). In more

severe meteorological conditions, when the polynya-averaged frazil concentration is relatively high, the streaks at some sections of the coastline (e.g., off the coast of Inexpressible Island) can be observed in the immediate vicinity of the shore (Fig. 4, Suppl. Fig. S2). In other areas, some of the streaks touch the border of the ice sheet. A typical example is the relatively wide streak which can be seen in most of the satellite images eastward off Cape Russell and extending from the coast to the downwind boundary of the polynya. The location of this streak also indicates an area where consolidation of

frazil and its transformation to other ice types are frequently observed, often resulting in separation of the small section of the polynya in the northern part of the TNB from its main part south of that area (Suppl. Fig. S1).

Starting from the location where the streaks first become visible, the surface area covered with frazil ice increases rapidly with increasing distance from the coast at the cost of the area of open water that separates them (Fig. 3b). Frazil concentration averaged in 500 m wide zones parallel to the shore ($C_Z$) reaches 50% about 1 to 3.5 km from the coast at more

severe weather conditions, and 4 to 8 km when the temperature is higher and the wind weaker. A further increase in concentration is slower. In proximity of the polynya border, regardless of its distance from the ice sheet, an increase in the concentration of frazil was observed. This increase has not been uniform along the entire length of the border, as can be seen on maps of spatially-averaged frazil concentration $C_S$ (Suppl. Fig. S3). In most cases, a greater frazil concentration occurs along the border closing the polynya from the north-east (e.g. Fig. 3c). The opposite situation with greater accumulation of

frazil near the ice bordering the polynya from the south to the south-east was also observed (e.g. Fig. 3d), but less frequently. In general, the curves describing changes of $C_Z$ with the distance from the ice sheet $X$ shown in Fig. 3b can be approximated by the function of the form:

$$C_Z(X) = 1 - \exp(X_T), \tag{9}$$

where $X_T$ is transformed distance $X$ (in km). The best fit results were obtained by transforming the distance according to

$$X_T = -t_1 X^{t_2}, \tag{10}$$

where $t_1$ and $t_2$ are the best fit coefficients. The stronger the wind, the better the fit (Fig. 5, Suppl. Fig. S4). Polynya events with the average wind speed higher than 25 m·s$^{-1}$ have an RMSE below 0.06. Larger residues occur when weather conditions are relatively mild and ice transformation can be observed within the polynya, or if consolidated ice made it difficult to delineate the polynya edge.

The best fit coefficients of distance transformation, $t_1$ and $t_2$, depend on weather conditions (Eq. 11 and 13 in Tab. 4, Fig. 6). The scaling factor $t_1$ is determined by the temperature change and the exponent $t_2$ by the wind speed and both can be approximated by simple linear regression (Eq. 12 and 14 in Tab. 4). Thus, for a given offshore distance $X$, $C_Z$ increases with decreasing air temperature and mostly decreases with increasing wind speed. The latter effect is related to the positive correlation between wind speed and the overall extent of the polynya. However, it can be compensated by the temperature

drop, due to the inverse correlation between the wind speed and the temperature which is noticeable for smaller distances



(less than 10 km). Although the relationships of $t_1$ and $t_2$ with meteorological parameters explain only about 50% of the coefficients' variance, they can be useful for predicting changes of frazil concentration with the distance (Fig. 6c). The RMSE error of the $C_Z$ estimate doubles when $t_1$ and $t_2$ are estimated from meteorological data, however, the Eq. 12 and 14 (Tab. 4) were derived from a small number of data, and the scatterplots (Fig. 6a and Fig. 6b) show that outliers may have influenced the regression results. It seems that taking into account more observations in the future may improve the parameterization of $t_1$ and $t_2$.

### 3.3. The width and orientation of frazil streaks

The spatial distribution of the lengths of cross-sections through the frazil streaks ($W_{FS}$) are shown in Suppl. Fig. S5. When considering the entire polynya, the distribution of frazil streaks' widths is strongly skewed with the dominant width of the streaks ranging from 20 to 100 m, regardless of the conditions in which the polynya was formed (Fig. 7a). In the analysed cases, the median and interquartile range of the streak widths change in the range of 40–60 m and 60–160 m , respectively. The area covered by frazil in narrow strips, the width of which does not exceed 100 m, constitutes 11–29% of the entire area of the frazil streaks in the polynya (Fig. 7b).

The narrowest streaks are present in the entire polynya, but are most abundant in the coastal zone (Fig. 8). The contrast between water and frazil in the satellite images for these narrow bands is relatively low, making detection difficult. Hence their strong fragmentation into short, straight sections. Further from the ice sheet, the narrowest streaks (10–50 m) still appear frequently in the spaces between the wider ones, which results in the widening of the distribution of streak widths (log-transformed) and even the appearance of a second mode. Near the downwind border of polynya, the accumulating frazil ice consolidates causing increase in width of streaks or the formation of wide patches of frazil/grease ice. The upper limit of the width which still referred to the distinct streaks was set to 3 km. The widest cross-sections, in some places up to 10 km, characterized areas of frazil accumulation. It is worth noting that even at the downwind border, very narrow streaks were numerous in the 'ponds' of water trapped between the patches of freezing frazil, as can be seen, for example, in the enlarged parts of the image in Fig. 1e-f.

Wider streaks are clearly visible in the central parts of the polynyas and at their downwind border (Fig. 8 and more examples in Suppl. Fig. S5). Although they are usually less numerous than the narrow ones, their contribution to the total surface area covered with frazil is much larger (Fig.7b). Streaks 100 m–1 km wide account for 31–73% and those 1–3 km wide for 6–35% of the frazil area in the polynya. A closer look at the spatial variability of width allows us to distinguish two types among them. The typical ones consist of streaks of medium width. These are streaks that often merge together and break apart again to form a kind of net. Locally, at junction points their cross-section increases to 1 km or more. On the other hand, at long distances, streaks maintain their width within the range of 200-300 m. In larger polynyas, a gradual increase in their width to 500–700 m is noticeable. The second type of wider streaks are those in the shape of the letter V, the width of which grows quickly to 1 km and more. They are characteristic of the border zone. In many of the analysed polynyas, however,



individual streaks of this type are observed, which extent from the border of the polynya far towards the ice sheet and even reach it, as mentioned before the streak eastward off Cape Russell (Fig. 8a–c).

Apart from the periphery areas of the polynya (near shore or downwind boundary), the water which separate frazil streaks is usually 30-150 m wide. The distribution of the distances between adjacent streaks is less skewed than the distribution of their width and can be approximated by the log-normal function with the parameters µ and σ in the range of 5.1–5.5 and 0.7–1.0, respectively. It means that median distances are in the range of 170–260 m. If the narrowest streaks with an average width of less than 50 m are omitted, the median is in the range of 270–440 m, which is more consistent with the range of 300–500 m

found by Ciappa and Pietranera (2013) on the basis of SAR data. The geometric standard deviation factor does not change significantly, being in the range of 2.0–2.6.

The orientation of the streaks vary in the range of angles from 60° to 160° (the angle, measured clockwise, between the North and the given segment of the streak skeleton oriented towards downwind edge of polynya). Most often they extend towards ENE to SE (histograms of the frazil streaks orientation for individual polynyas are shown in Suppl. Fig. S6). The

winds in the analysed situations usually blew in the direction ENE to E, with the exception of one outlier situation on September 20, 2012, when the wind towards SE has been observed (Tab. 2). In this case, the mean direction of the streaks is close to the wind direction measured just before satellite overpass, deviating only slightly to the left (−6.5°). In all remaining cases in which the wind direction at Manuela station is known, the mean direction of the streaks is deflected 10° to 30° to the right of the wind direction and there is strong correlation between them ($r$=0.86, *p-level*=0.0000).

The spatial distribution of the frazil streaks orientation indicates the highest azimuthal angles of the bands in the area south of the Inexpressible Island (Fig. 9 and more examples in Suppl. Fig. S7). In this part of the bay, the orientation of streaks in the coastal zone is usually perpendicular to the boundary line of the ice sheet or tilted to the right, and most streaks maintain their general orientation further from the shore. In the northern part of TNB, despite the similar orientation of the coastline, the azimuth of the streaks is smaller by 10–50 degrees than in the southern part. They extend generally to the east, changing

their inclination towards northeast with the distance from the coast and towards north of the TNB. In the polynyas with a large extension along the ice sheet and a long range from the shore, the fan-shaped pattern of the streaks can be clearly visible. The spatial variability of streak orientation is different in the case of the streaks with greater accumulation of frazil reaching the polynya border. They changed their orientation locally by 10 degrees or more, in both directions (e.g. Fig. 9c, f).  Figure 10 shows orientations of the frazil streaks identified in all analysed polynyas, divided into four size classes, and

the corresponding wind directions. It should be noticed that the variability in the orientation of the widest streaks (an average of 1−3 km along the skeleton section) is the highest, with typical directions in the range of ENE−SSW, and the correlation with the wind direction is relatively weak ($r$=0.52, *p-level* =0.0095). The most stable in terms of orientation are the narrowest streaks (on average less than 100 m wide) with dominant directions ESE−SE. The relationship with the wind direction is also the strongest in this case ($r$=0.87, *p-level* =0.0000).



### 3.4. Waves

The curves of $L_{p,n}(X_n)$ for the 18 polynya cases for which they could be robustly determined (see Section 2) are shown in Fig. 11 together with the corresponding open-water curve from SWAN. The data have been divided into three wind-speed classes to illustrate the role of that factor (and the correlated polynya extent) in the wave growth. At moderate wind speeds, below 17 m·s⁻¹, the computed values of $L_p$ tend to be comparable or even higher than the corresponding $L_{p,OW}$, especially within the first 10–15 km from the coast (Fig. 11b). Accordingly, the $L_{p,n}(X_n)$ curves have very mild slopes. However, as the values of $L_p$ in those cases lie within the 30–40 m range, it is likely that this behaviour can be at least partly attributed to uncertainties in wavelength estimation. For higher wind speeds, above 17 m·s⁻¹, the agreement between $L_p$ and $L_{p,OW}$ tends to be very good within the first few kilometers from the coast, but the discrepancy between them increases with both $X$ and $U_w$: the stronger the wind and the larger the fetch, the smaller the ratio between $L_p$ and $L_{p,OW}$. In the largest polynyas, extending to over 40 km from shore, the observed peak wavelength is never larger than 60–70 % of the expected open-water wavelength. Possible causes for that behaviour are discussed in the next section.

The mean wave directions at the peak wavelength, obtained from the Fourier analysis of these 18 polynya events, are in the range of angles from 60° to 120°, i.e. ENE to ESE directions. The spatial distribution of wave directions (Fig. 12a–b and more examples in Suppl. Fig. S8), despite the smaller variability of the angles, shows a similar pattern as in the case of the frazil streaks orientations, described earlier (Fig. 9e–f and more examples in Suppl. Fig. S7), i.e. smaller angles in the northern part of TNB (ENE to E) and larger to the south (E to ESE). Due to the erroneous results of the Fourier analysis in the coastal zone and the frequent lack of distinct frazil streaks in the border zone, a more detailed comparison of the mean orientation of streaks and wave directions is possible only for the largest polynyas. The relationship between $\theta_p$ and $\theta_{FS}$ may be approximated by the logarithmic function (Fig. 12c–d), which very well illustrates the smaller range of variability of wave directions in regards to the frazil streaks orientation. The difference of angles reaches 30° and is larger in areas where the frazil streaks are tilted towards the south-east (Fig. 13).

### 4 Discussion and conclusions

Coastal polynyas are very dynamic, constantly changing environments shaped by the local atmospheric forcing as well as the surrounding sea ice and oceanic conditions (extent, thickness, compactness and motion patterns of the ice pack, local and regional ocean currents). The available satellite and other data show that the shape and extent of polynyas evolve on a daily or even hourly basis (Kern et al. 2007, Ciappa and Pietranera, 2013, Aulicino et al., 2018), indicating that processes taking place there are nonstationary and very sensitive to changes in the forcing. The images analysed here and in similar, satellite-based studies are single snapshots of those evolving systems – or, more precisely, snapshots limited to the ocean surface. As argued in the introduction, patterns of frazil streaks visible in those snapshots are indicators of dynamic processes taking place in the OML.



Despite the limited number of available scenes recorded with high-resolution optical sensors, the analysed data set contains diverse cases in terms of the shape and area of the polynyas. Most of the scenes cover the entire area of the open water adjacent to the Nansen Ice Sheet. Our analysis ignores openings along and at the extremity of the Drygalski Ice Tongue, as well as any leads external to the polynya, hence the estimated areas ($S_p$) may be smaller than those estimated on the basis of

thermal data for the entire TNB by Ciappa and Pietranera (2012) or Aulicino et al. (2018). In this study, the analysis was also limited to the area of the polynya where frazil streaks are visible. Ciappa and Pietranera (2013) describe the presence of separated ice bands reaching up to the downwind border as a feature of polynyas in the opening phase. In this phase, a narrow zone of accumulating frazil ice at the edge of consolidated ice designates edge of the polynya. In the analysed cases, such a rim is clearly visible usually only in some parts of the polynya. In the remaining parts the border is blurred into a

broad zone of frazil accumulation and transformation. Considering the high dynamics of the formation and disappearance of polynyas described in previous studies (Kern et al. 2007, Ciappa and Pietranera, 2012, Aulicino et al., 2018), it is expected that alternating phases of those processes often lead to such an uneven distribution of the frazil at the surface of the polynya.

The results presented in previous sections have shown two important facts. First, several features of frazil streaks (changes of area with the distance from the ice sheet, the distribution of the width and orientation of the streaks, and the tendency to form

narrow bands along the entire width of the polynya wherever open water 'ponds' appear) exhibit high repeatability in spite of the overall strong variability of polynya sizes and shapes, illustrating certain universal mechanisms governing the development of streaks and the overall rates of frazil production and transport. And second, a substantial part of the observed variability can be linked to the local atmospheric forcing. In spite of the relatively large uncertainty for individual cases, the empirical relationship between air temperature and wind speed at a single point onshore, and increase of frazil concentration

with offshore distance is robust and provides a useful practical tool that can be applied, among other things, to improve predictions of the net ocean–atmosphere heat transfer and ice production.

The existing estimates of these quantities (e.g., Nakata et al., 2021 and references there) are based on low-resolution satellite products, from which no information on the open-water and frazil/grease-ice-covered surface can be obtained, crucial for heat flux computation (Guest. 2021b, Ackley et al., 2022). In this context, high-resolution radar-based imagery from SAR

systems seems a very attractive data source for subsequent studies, as it would enable extending the analysed dataset to polynya events in clouded conditions and during the polar night, i.e., situations when no visible imagery is available. It cannot be ruled out that the correlations found in this study are weaker during the winter months, when the more extensive and compact ice pack limits polynya growth stronger than in the spring months of September and October, considered here. However, the importance of the available data recorded in the visible spectrum with radiometers such as MSI on Sentinel-2

or OLI on Landsat 8 and 9, which provide high spatial resolution data (10–15 m) over a relatively wide swath (185-290 km) should be emphasised. Earlier studies, based on high-resolution (5 m) SAR data, characterize frazil ice appearing 3–4 km from the ice sheet, forming streaks separated by 300–500 m (Ciappa and Pietranera, 2013). The results described in Section 3.2 indicate that visible data enables detection of relatively narrow streaks less than 100 m wide, which on average contribute 20% to the total surface covered with frazil. Those narrow streaks undetected by SAR occur close to the shore, as



well as in open water spaces between wider bands of accumulated frazil up to the downwind border of the polynya. Very high correlation of their orientation with the wind direction ($r = 0.87$) suggests that their spatial distribution may reflect the properties of Langmuir cells. Moreover, high resolution SAR modes, up to 5 m per pixel, usually cover swath less than 100 km wide. The typical wide swath SAR recording mode, which is able to cover the entire TNB polynya region, provides data with a lower resolution of 30–100 m per pixel. The detection of narrow bands is additionally hampered by the speckling

effect. In order to better use the synergy of data registered with different techniques, effort must be put into the development of the classification methods to distinguish frazil streaks from water and other types of ice, and inter-calibration of the results.

    A very interesting aspect of the results is the influence of wind speed on the $C_Z(X)$ curves, showing a slower increase of ice concentration with distance with increasing $U_w$. The wind speed influences $C_Z(X)$ by modifying (i) the drift speed of the ice

(and thus the polynya extent), (ii) the distribution of the ice crystals within the water column, and (iii) the surface heat flux (and thus the ice production rates). The net slower increase of surface ice concentration with increasing wind speed suggests that the dynamic effects (fast drift, possibly combined with strong mixing) dominate over thermodynamic ones. This finding is in agreement with the recent results by Ohshima et al. (2022), who found a strong positive correlation between wind speed and penetration depth of frazil in a coastal polynya in the Eastern Antarctic. As Fig. 3b shows, only when the $C_Z(X)$ curves

are scaled with polynya extent the role of higher freezing rates at higher wind speeds becomes visible. Notably, mixing of frazil down the water column additionally enhances the surface heat flux by delaying the formation of a continuous grease ice layer at the surface.

    In a wider perspective, *in situ* observations and very-high-resolution, coupled wave, current and ice models of polynyas, nested in regional models, are essential for extending the present, surface-based view to a full three-dimensional (3D)

picture. Idealised numerical models (e.g., Herman et al., 2020) clearly show complex 3D patterns of ice concentration related to the OML dynamics. At the same time, recent high-resolution modelling in the TNB basin provides new insights into its 3D circulation, with OML in front of the Nansen Ice Sheet influenced by circulation under the ice sheet (Na et al., 2022). Although the presence of submesoscale eddies in TNB during katabatic wind events have been analysed only during summertime (Moctezuma-Flores et al., 2017, Friedrichs et al., 2022), it seems reasonable to assume that at least some

elements of that circulation are present during polynya events as well. That would explain the fan-like frazil streak patterns described in this study, with a tendency to anticyclonic/cyclonic rotation in the northern/southern part of the polynya. Similarly, the presence of very stable streaks originating at the headlands, island tips and other protruding points at the coastline indicate that they are associated with convergence zones of local circulation cells. As said, a better understanding of those features and relationships requires observational and modelling insights that are lacking at present.

Overall, as mentioned in the previous section, both peak wave directions and streak directions reveal a repeatable fan-like pattern, with much narrower directional variability of waves than that of streaks and increasing variability with increasing frazil accumulation. Due to a slight asymmetry in the $\theta_{FS}$–$\theta_p$ relationship (Fig. 12c,d and 13), frazil streaks tend to be oriented along the peak wave direction in the northern parts of the polynyas (for $\theta_{FS}$ below 90°) and to the right from the



peak wave direction in the remaining parts, with turning angles increasing to the south. In a subsequent study, a two-
dimensional setup of the SWAN model, forced with realistic, spatially variable wind patterns, will be used to analyse the
relationships between the local wind direction, peak and mean wave direction, and Stokes drift direction in different parts of
the polynyas.

A very important question related to the observed patterns of streak and wave directions is how they are related to the local
wind forcing. Answering this question is nontrivial with the available data. As discussed in Section 3.3, and as can be seen in
Table 2, Fig. 10 and Suppl. Fig. S6, almost all wind directions measured at the Manuela station during the analysed polynya
events are from the WSW–W sector  (see also Friedrichs et al., 2022), and the streaks are oriented primarily to the ESE, that
is, to the right of the wind. This relationship holds not only when analysed globally (i.e., over the whole dataset, or over the
entire surface of a given polynya, as in Suppl. Fig. S6), but also locally: the streaks in the direct vicinity of the Inexpressible
Island are oriented to the right of the wind measured at Manuela (see maps in Suppl. Fig. S7). Assuming that the thin,
nearshore streaks reflect the surface part of the local Langmuir circulation, which in turn can be expected to follow the
direction of locally generated wind waves, this would suggest a right angle between the wind at Manuela and the wind over
water directly offshore. (Recall that, unfortunately, the peak wave directions there cannot be reliably estimated due to too
small wavelengths.) Most probably this effect is related, first, to the fact that the AWS is located at the height of 78 m a.s.l.,
and second, to some orographic influences, impossible to quantify without other, independent data sources. Importantly,
however, even if those effects are present, they seem to produce a bias in the measured wind directions and to narrow the
range of observed values without disturbing their overall variability. Otherwise, it would be hard to explain the remarkably
high correlation, exceeding 0.8, between the measured $U_w$ and the mean streak direction for individual polynyas, Notably,
that correlation is even higher, close to 0.9, if the only streaks less than 100 m in width are taking into account. In short, the
available data shows a very strong relationship between the direction of the wind forcing and the observed peak wave
directions and frazil streak orientation, even though the obtained mean angles are presumably biased. Thus, at least for the
large polynyas, the overall spatial pattern of both frazil streaks and wave directions can be predicted from Manuela
observations.

An alternative approach to the problem of wind data would be to use wind fields from a regional atmospheric model, like
e.g. the Antarctic Mesoscale Prediction System (AMPS; Powers et al., 2003, 2012). Indeed, our preliminary analysis of
AMPS data has shown that the AMPS wind directions at and downwind from the Inexpressible Island are oriented to the
ESE and agree well with the observed streak and wave directions in that area. However, the setup and spatial resolution of
the AMPS model changed twice in the period of interest (from 15 km to 1.1 km, and then 0.9 km), and those changes are
clearly visible in the results. In particular, the modelled wind speeds during polynya events in the years 2009 and 2012 are
markedly lower than those in the period from 2016 onward, and the correlations between $U_w$ and polynya area, extent and
frazil concentration, reported in this paper, get much worse when wind speeds from Manuela are replaced with those from
AMPS. How reliable the modelled wind directions (directly at the coast and further offshore) are in different AMPS versions
is therefore very difficult to estimate, and such an analysis is beyond the scope of this study. (Intriguingly, Wenta and





Cassano, 2020, found a very good correlation between the Manuela and AMPS wind speeds; however, their analysis was limited to a short period in September 2012, i.e., the only situation in our dataset when the Manuela wind was from the NW

sector.)

As mentioned in Section 2, the type of imagery used in this study does not allow estimation of wave heights. Similarly, although the very different character of the sea surface within the open water and grease ice 'patches' can be seen with the naked eye, it is that 'patchiness' that makes the analysis of wave properties challenging. In particular, performing FFT on masked images in order to obtain separate spectra for ice and open water produces ambiguous results because FFT is very

sensitive to gaps in the data. In some cases, the presence of (quasi-periodic) frazil streaks leads to artificial peaks in the resulting spectra. Without *in situ* data that can be used to verify the results, making any inferences from the higher-frequency parts of the spectra is problematic. Nevertheless, interesting and important conclusions regarding wind wave development in coastal polynyas can be drawn from the quantities that are not affected by the aforementioned drawbacks: as $L_p$ and $\theta_p$ are largely unaffected by the presence of the ice, they can be robustly determined and, as described in the previous section, they

tend to significantly deviate from the corresponding open-water values. One of the possible reasons is that the actual conditions over polynyas do not correspond to those assumed in the idealized model. In particular, the wind speed measured at the Manuela station might not be representative for winds further offshore. To test whether that factor alone might explain the observed slowdown in the wave growth with fetch, we performed additional simulations with SWAN, similar to those described in Section 2, but with $U_w = U_{w,0}\exp(-X/s)$, where $U_{w,0}$ denotes the wind speed at $X = 0$ and $s$ is the length scale

describing the rate of decrease of $U_w$ with distance. The goal was to find values of $s$ necessary to reproduce the observed $L_p(X)$ curves. For the large polynyas with extent exceeding 40 km, $s$ as low as 25–30 km were needed, several times lower than the $e$-folding length scales known from observations during polynya events over TNB, exceeding 100 km and in some cases as large as 190 km (Guest, 2021b). With $s > 100$ km, ratios of $L_p/L_{p,OW}$ are never lower than ~80 %. Therefore, the role of decreasing wind speeds in producing the observed slowdown of wave growth is rather limited. Similarly, gradients of

wind and wave conditions in the across-wind direction cannot explain the observed data, as the wave conditions in the selected images are very uniform in that direction within a zone which is often a few tens of kilometers wide. Consequently, the presence of frazil/grease ice and its interactions with OML dynamics likely are the main cause of the observed wavelength variability.

**Code/data availability**

All satellite data used in this study is publically available at USGS Service (EO-1 and Landsat data) https://earthexplorer.usgs.gov/ and Copernicus Open Access Hub (Sentinel-2) https://scihub.copernicus.eu/. The code of the SWAN model can be downloaded from https://swanmodel.sourceforge.io/.



**Author contribution**

A.H. conceptualized the study, performed the Fourier analysis and spectral wave modelling. K.B. did all other data
processing and analysis, as well as visualisation of the results. Both authors discussed the results and wrote the paper.

**Competing interests**

The authors declare that they have no conflict of interest.

**Acknowledgements**

This work has been financed by Polish National Science Centre project no. 2018/31/B/ST10/00195 ("Observations and
modeling of sea ice interactions with the atmospheric and oceanic boundary layers"). Part of the data analysis was carried
out at the Academic Computer Center (TASK) in Gdańsk, Poland. A.H. is grateful to Stephen Ackley and Peter Guest for
discussions on polynya processes and preliminary results of this paper.

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





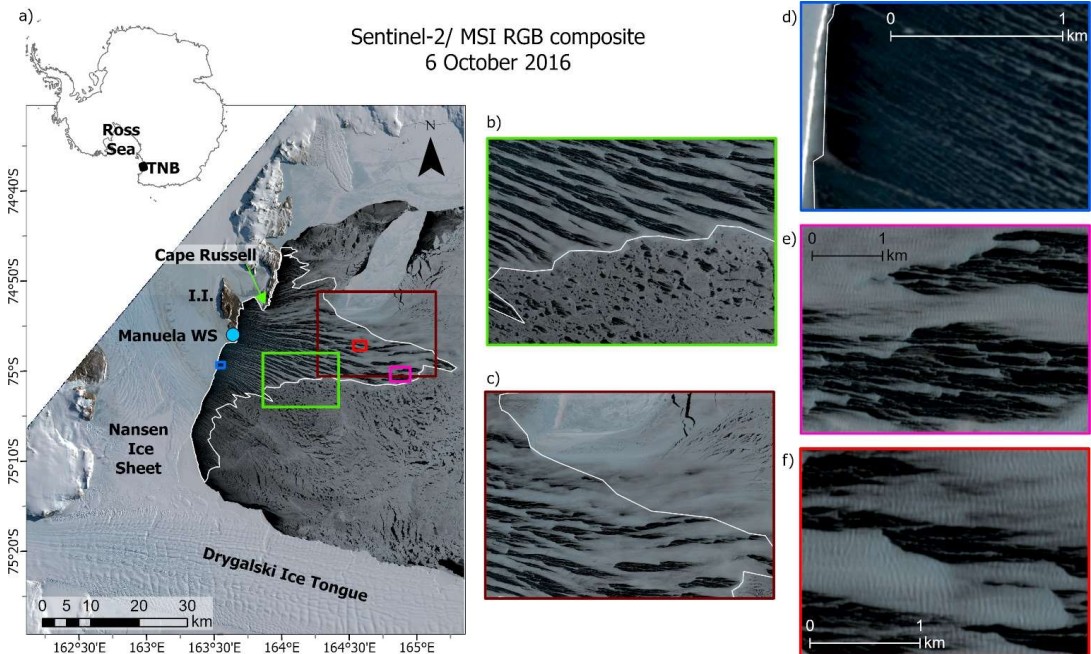

**Figure 1: The Terra Nova Bay (TNB) surroundings and an example of a polynya captured by Sentinel-2 (a); the outline of the polynya and the location of the Manuela Weather Station on the Inexpressible Island (I.I.) are marked with the white polygon and blue dot, respectively. The selected parts of the satellite image are enlarged to show: the edge of the polynya with distinct (b) and blended (c) border between frazil streaks and consolidated ice; narrow streaks of frazil near the ice sheet (d) or in the 'ponds' of water between wider streaks (e); as well as waves propagating in sea ice (f).**




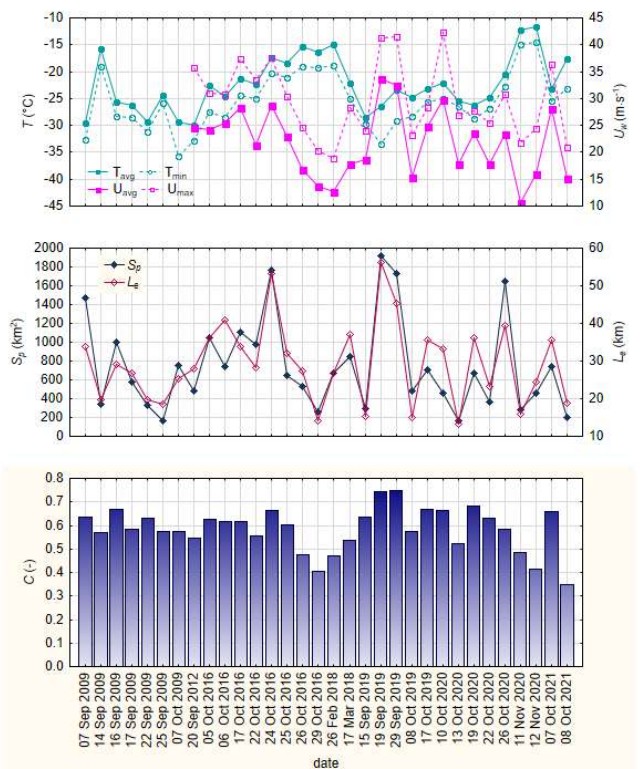

**Figure 2: Variability of meteorological parameters ($T$- temperature, $U_w$ – wind speed), derived from 10-minute observations at Manuela weather station, recorded during 24h preceding satellite overpass, and characteristics of the analysed polynyas: area ($Sp$), maximum extent ($L_e$) and frazil concentration ($C$) estimated from satellite data (different time intervals along $x$ axis).**





**Figure 3: Statistical characteristics of the distance from the ice sheet to frazil streaks ($X_{FS}$) (a), variability in the zone-averaged frazil concentration ($C_Z$) (500 m wide zones parallel to the shore) with the distance from the ice sheet ($X$) normalized to the maximum range of the polynya ($L_e$); the 24 hour mean wind speed ($U_w$) observed at Manuela station is shown by colour (b), and examples of spatial variability of frazil concentration averaged in 2.5×2.5 km moving window ($C_S$): 5 Oct 2016 (c) and 24 Oct 2016 (d).**





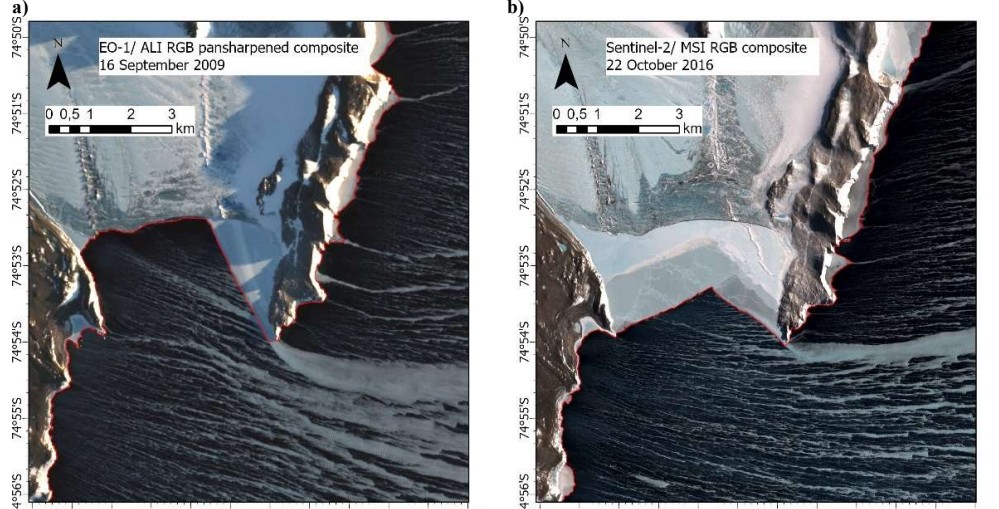

**Figure 4: Examples of frazil streaks visible in the immediate vicinity of the shore.**

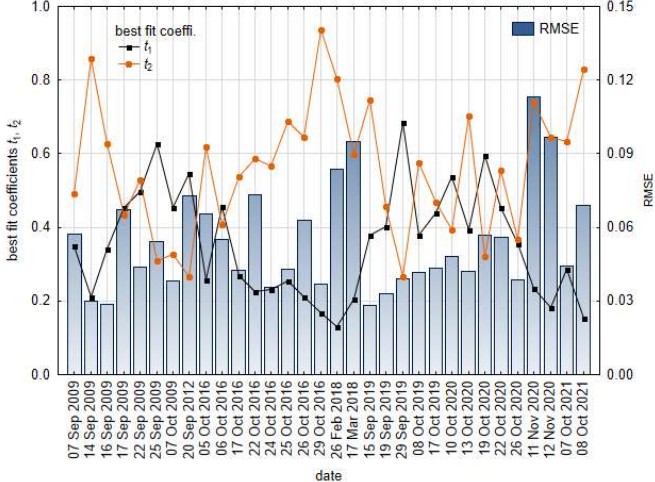

**Figure 5: The results of the approximation of the relationship between the frazil concentration and the distance from the ice sheet by Eqs. (9–10): best fit coefficients ($t_1$ and $t_2$) and root mean square errors (RMSE).**



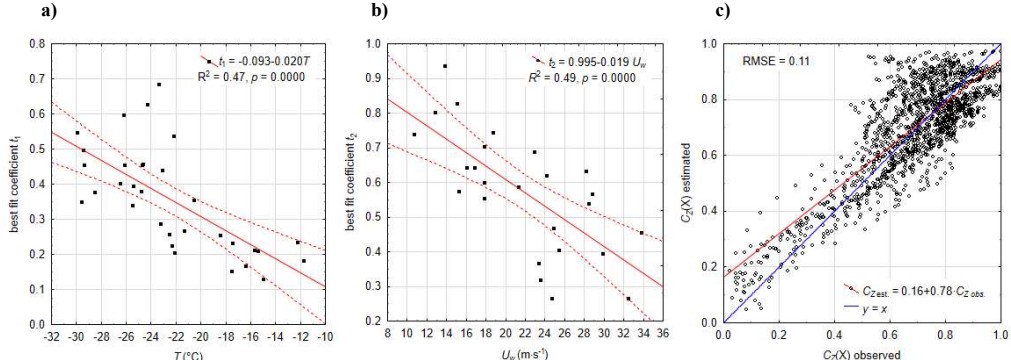

**Figure 6: Dependence of coefficient $t_1$ on temperature (a), coefficient $t_2$ on wind speed (b) (weather conditions averaged over 24h), with the corresponding relationship between the observed and estimated $C_Z(X)$ (c). Dotted lines in (a,b) mark the corresponding 95% confidence interval.**

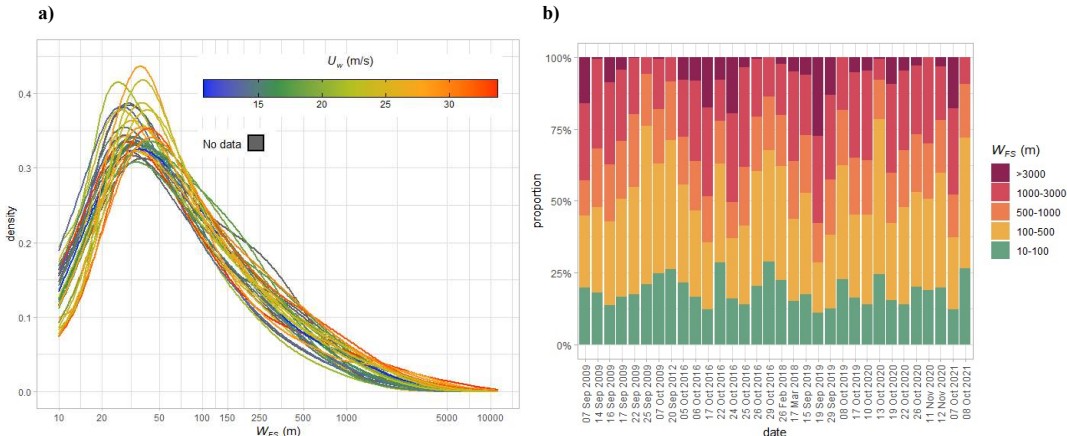

**Figure 7: Density distributions of the frazil streaks width ($W_{FS}$) for each polynya (a) and proportion of the area covered with segments of frazil streaks in five width classes (b); $U_w$ - average wind speed as observed at Manuela station during preceding 24h; a logarithmic scale was used for the streak widths to calculate density distribution.**



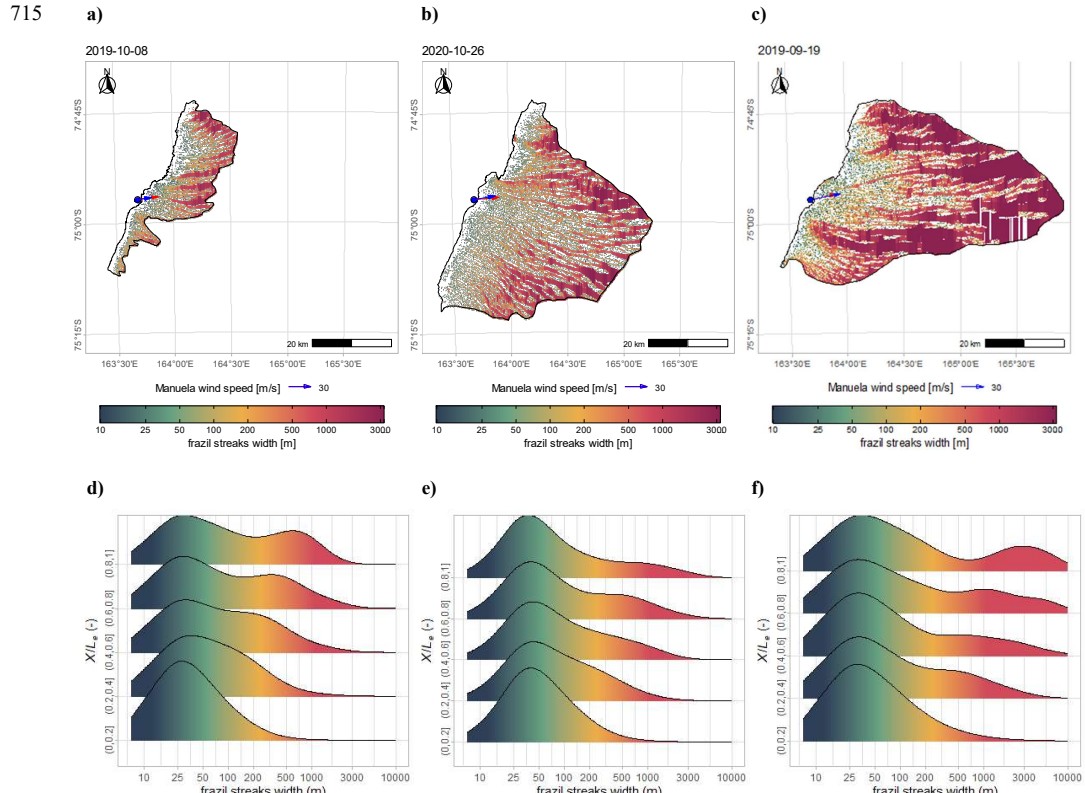

**Figure 8: Spatial variability of the frazil streaks width ($W_{FS}$) (a-c) and its density distributions at different zones (distance from the ice sheet $X$ normalized to the maximum range of the polynya $L_e$) (d-e) for selected polynya formed in various meteorological conditions: 8 Oct 2019 – $U_w$=15.3 m·s⁻¹ (a,d), 26 Oct 2020 – $U_w$= 23.3 m·s⁻¹ (b,e) and 19 Sep 2019 - $U_w$=33.8 m·s⁻¹ (c,f); a logarithmic scale was used for the streaks width, arrows indicate the wind direction measured at the Manuela station (red - instantaneous, blue - averaged over 24 h).**





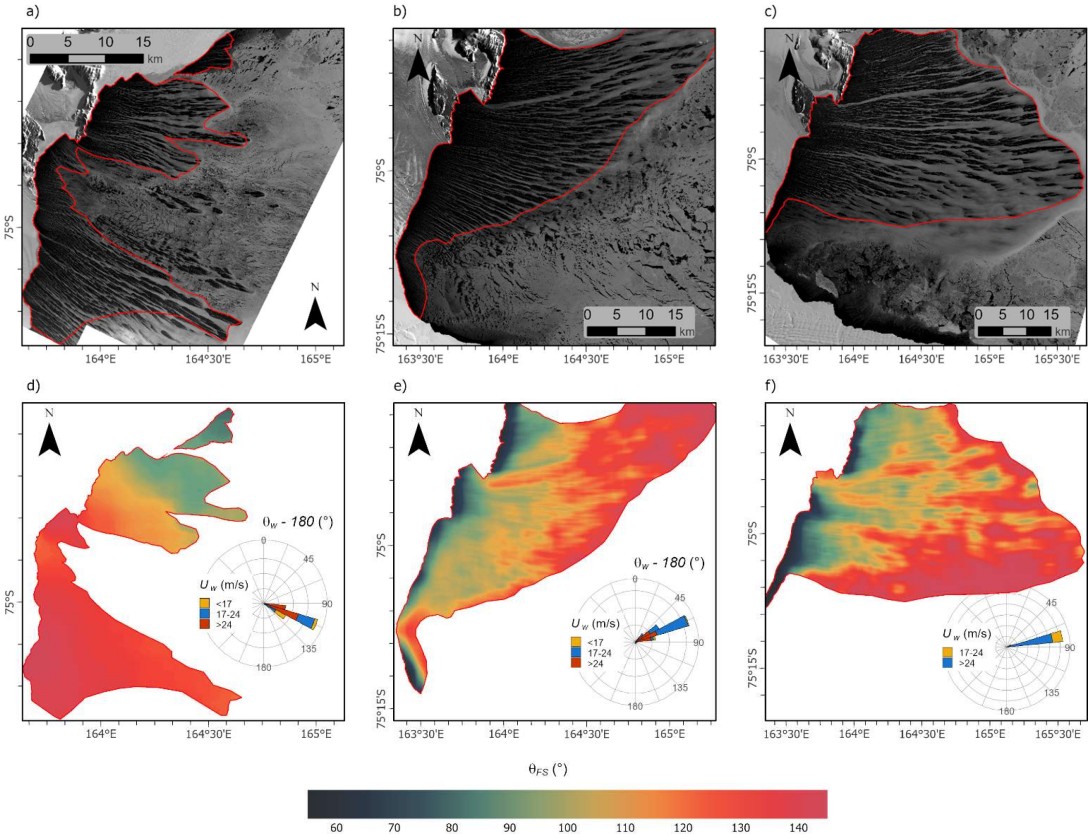

**Figure 9: Orientation of the frazil streaks visible on panchromatic images (a-c) and their orientation (θ$_{FS}$) averaged in 5.1×5.1 km**
730 **moving windows (512×512 pixels) (d-f) for selected polynyas: 20 Sep 2012 (a,d), 5 Oct 2016 (b,e) and 24 Oct 2016 (c,f); wind roses show speed and direction to which the wind blows (θ$_w$–180°; data measured at Manuela weather station for 24h before satellite image registration); erroneously determined angles in the area with no distinct streaks are marked by dashed polygons.**





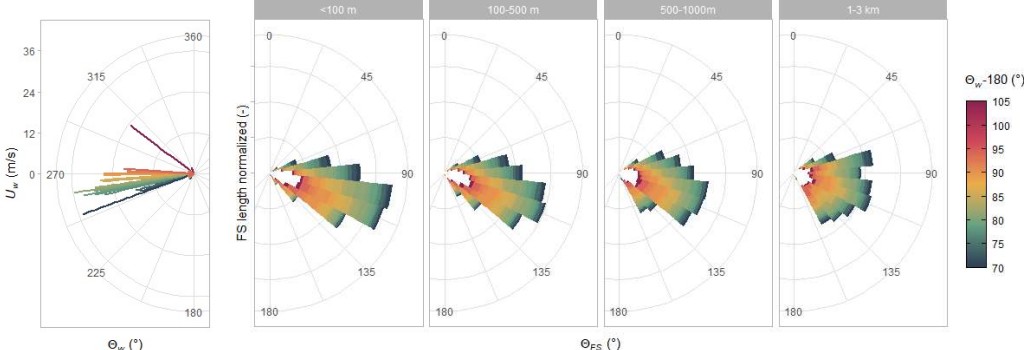

**Figure 10: Orientation of frazil streaks (θ$_{FS}$) in four classes of their size (average width along the skeletal section): less than 100 m, 100 – 500 m, 500 m – 1 km, 1 – 3 km, depending on the wind direction (θ$_w$) measured at the Manuela station just before satellite overpass, for all polynyas from the analysed dataset; box length means the total length of skeletal sections of streaks oriented in a given direction, normalized with the total length in all directions in the analysed polynya for each size class, the colour scale indicates the direction in which the wind was blowing (θ$_w$−180) - white boxes represent data from the year 2009, from which wind data were not available; in addition, the speed (U$_w$) and direction of the wind are indicated by arrows in the left panel.**

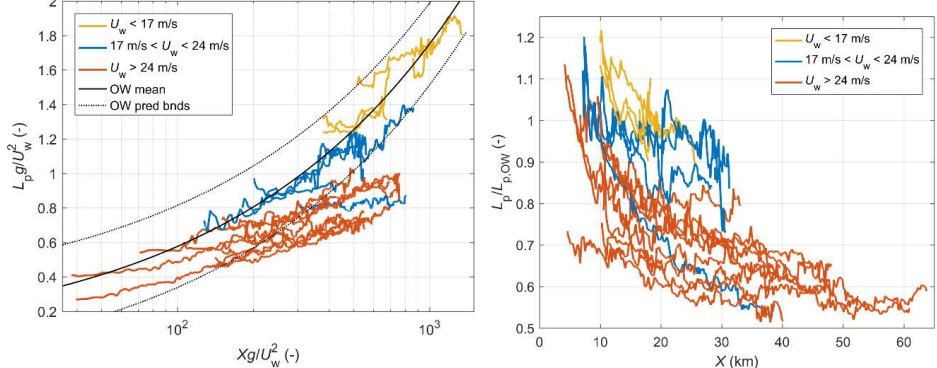

**Figure 11: Peak wavelength $L_p$ obtained from the Fourier analysis of the polynya imagery: dimensionless wavelength $L_p g/U_w^2$ in function of dimensionless fetch $Xg/U_w^2$ (a) and relative wavelength $L_p/L_{p,OW}$ in function of fetch $X$ (b). The polynyas are divided into three wind speed classes (colors). In (a), continuous black line shows the mean open-water wavelength $L_{p,OW}$ obtained with SWAN simulations with various wind speeds and source term combinations (see text for details); dotted black lines mark the corresponding 95% confidence interval of $L_{p,OW}$.**



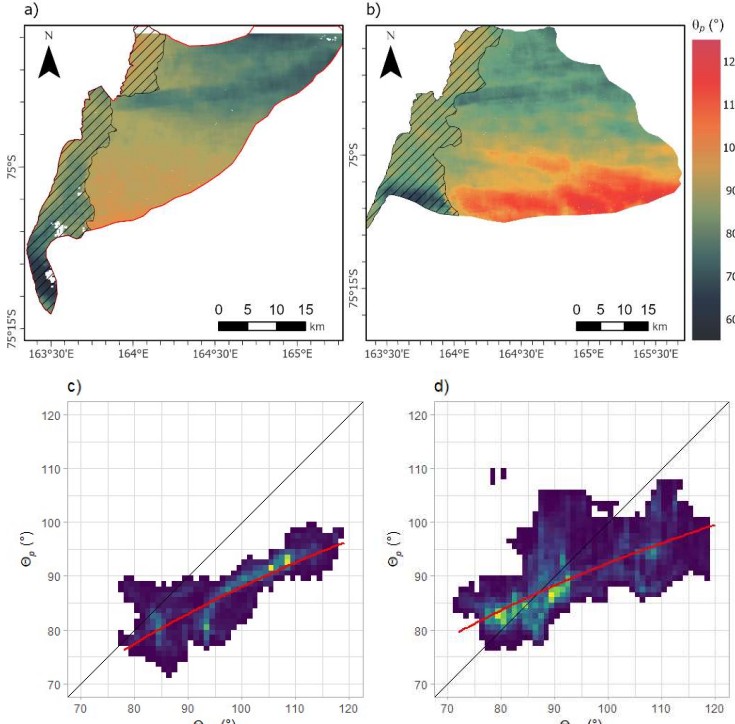

**Figure 12: Spatial distribution of mean wave direction at the peak wavelength θ$_p$ obtained from the Fourier analysis of the polynya imagery (a-b) and relation between mean direction of waves and frazil streaks (c-d) for examples of large polynya 5 Oct 2016 (a,c) and 24 Oct 2016 (b,d); the uncertain results of the Fourier analysis are marked by dashed polygons and omitted in scatterplots; red line in scatterplots indicates fit by the logarithmic function.**



755

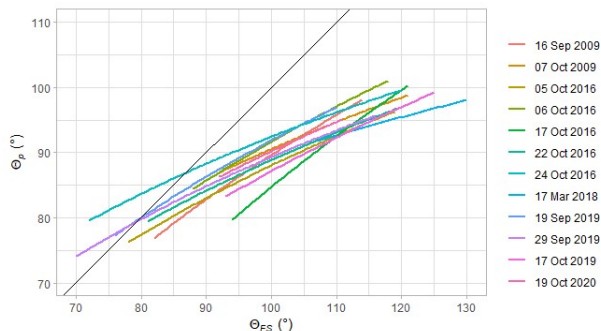

**Figure 13: A comparison of logarithmic fits of the relationship between mean direction of waves $\theta_p$ obtained from the Fourier analysis and mean frazil streaks orientation $\theta_{FS}$ for large polynyas; black line indicates $y = x$.**

760    **Table 1. Characteristics of satellite data used in this study**

| satellite / sensor | band | spectral range (μm) | spatial resolution (m) | swath width (km) | product type and projection |
|---|---|---|---|---|---|
| EO-1 / ALI | PAN | 0.480 – 0.690 | 10 | 37 | L1GST WGS84/UTM58S |
| | MS – 5 | 1.550 – 1.750 | 30 | | |
| Landsat-8 / OLI | PAN | 0.500 – 0.680 | 15 | 185 | L1GT (C2) WGS84/Polar Stereographic |
| | SWIR 1 | 1.566 – 1.651 | 30 | | |
| Sentinel-2 / MSI | Blue | 0.458 – 0.523 | 10 | 290 | S2MSI1C WGS84/UTM58S |
| | Green | 0.543 – 0.578 | 10 | | |
| | Red | 0.650 – 0.680 | 10 | | |
| | SWIR | 1.565 – 1.655 | 20 | | |



**Table 2. Characteristics of analysed scenes, meteorological conditions at Manuela weather station: *T* – air temperature, *Uw* – wind speed (averaged over 24h before satellite overpass), θ$_w$ - the direction from which the wind blows (measured just before satellite overpass) and size of polynya: *S$_p$* – area, *L$_e$* – maximal extent from the ice sheet, *L$_c$* – extent along the coast**

| date | Time (UTC) | sensor | $T$ (°C) | $U_w$ (m·s$^{-1}$) | θ$_w$ (°) | $S_p$ (km$^2$) | $L_e$ (km) | $L_c$ (km) | |
|---|---|---|---|---|---|---|---|---|---|
| 7 Sep 2009 | 2110 | ALI | -29.6 | | | 1474 | 34.0 | 64.5 | [1,2] |
| 14 Sep 2009 | 2040 | ALI | -15.7 | | | 347 | 19.6 | 39.5 | |
| 16 Sep 2009 | 2020 | ALI | -25.5 | | | 1003 | 29.2 | 79.7 | [1] |
| 17 Sep 2009 | 2100 | ALI | -26.2 | | | 576 | 26.7 | 71.6 | |
| 22 Sep 2009 | 2100 | ALI | -29.4 | | | 334 | 19.6 | 43.4 | [2] |
| 25 Sep 2009 | 2110 | ALI | -24.3 | | | 161 | 18.5 | 28.4 | |
| 7 Oct 2009 | 2040 | ALI | -29.4 | | | 753 | 25.4 | 51.5 | [1] |
| 20 Sep 2012 | 2110 | ALI | -29.9 | 24.7 | 307 | 483 | 27.9 | 53.2 | |
| 5 Oct 2016 | 2120 | MSI | -22.5 | 24.1 | 260 | 1043 | 36.2 | 63.7 | |
| 6 Oct 2016 | 2050 | MSI | -24.6 | 25.4 | 262 | 740 | 40.8 | 62.3 | [3] |
| 17 Oct 2016 | 2050 | OLI | -21.4 | 28.4 | 261 | 1110 | 33.8 | 46.7 | [3] |
| 22 Oct 2016 | 2110 | MSI | -22.3 | 21.3 | 259 | 975 | 28.3 | 46.8 | [3] |
| 24 Oct 2016 | 2100 | OLI | -17.4 | 28.7 | 257 | 1762 | 53.3 | 55.2 | |
| 25 Oct 2016 | 2120 | MSI | -18.4 | 22.9 | 270 | 647 | 32.1 | 53.0 | |
| 26 Oct 2016 | 2050 | MSI | -15.4 | 16.8 | 266 | 528 | 27.4 | 49.1 | |
| 29 Oct 2016 | 2100 | MSI | -16.4 | 13.8 | 252 | 260 | 14.1 | 41.8 | |
| 26 Feb 2018 | 2100 | MSI | -15.0 | 12.8 | 254 | 676 | 26.9 | 57.0 | |
| 17 Mar 2018 | 2130 | MSI | -22.1 | 17.8 | 270 | 850 | 37.0 | 58.7 | |
| 15 Sep 2019 | 2100 | OLI | -28.6 | 18.7 | 265 | 289 | 15.3 | 35.3 | |
| 19 Sep 2019 | 2100 | MSI | -26.5 | 33.8 | 258 | 1920 | 56.3 | 50.0 | |
| 29 Sep 2019 | 2110 | OLI | -23.4 | 32.4 | 250 | 1729 | 45.4 | 57.9 | [2,3] |
| 8 Oct 2019 | 2130 | MSI | -24.8 | 15.3 | 262 | 480 | 15.1 | 50.4 | |
| 17 Oct 2019 | 2100 | OLI | -23.1 | 24.9 | 265 | 707 | 35.5 | 47.6 | |
| 10 Oct 2020 | 2100 | OLI | -22.2 | 29.9 | 269 | 463 | 33.1 | 38.2 | |
| 13 Oct 2020 | 2100 | MSI | -25.4 | 17.9 | 274 | 167 | 13.3 | 25.9 | [2] |
| 19 Oct 2020 | 2100 | OLI | -26.2 | 23.5 | 261 | 674 | 36.2 | 46.9 | |
| 22 Oct 2020 | 2130 | MSI | -24.7 | 17.8 | 270 | 362 | 23.2 | 49.3 | |
| 26 Oct 2020 | 2100 | OLI | -20.6 | 23.3 | 266 | 1648 | 39.5 | 65.7 | |
| 11 Nov 2020 | 2130 | MSI | -12.3 | 10.7 | 263 | 286 | 15.8 | 34.3 | |
| 12 Nov 2020 | 2100 | MSI | -11.7 | 16.0 | 271 | 464 | 24.3 | 62.5 | |
| 7 Oct 2021 | 2130 | MSI | -23.2 | 28.1 | 272 | 736 | 35.5 | 52.2 | |
| 8 Oct 2021 | 2100 | MSI | -17.5 | 15.1 | 274 | 195 | 18.9 | 35.8 | |

[1] situations where polynya extends over the edge of the available image
[2] situations where polynya is partially clouded
[3] classification scheme adjusted





770   Table 3. Statistical characteristics of multiple and simple regression for estimating parameters of polynyas: area ($S_p$), extent ($L_e$), frazil concentration ($C$) and median distance from ice sheet to frazil streaks ($Md(X_{FS})$) on the basis of meteorological conditions: wind speed ($U_w$) and temperature ($T$) in 24h period preceding satellite overpass. For multiple regression partial correlation coefficients and corrected determination coefficients ($R_c^2$) are given, simple regression is determined for the selected independent variable with a higher correlation coefficient, ignoring the influence of the second one; coefficients

775   statistically significant at $p<0.05$ are bolded; for regression coefficients standard errors of estimation are given in parentheses.

| dependent variable | regression | correlation coefficients | | regression equation | regression statistics |
|---|---|---|---|---|---|
| | | $T$ (°C) | $U_w$ (m·s⁻¹) | | |
| $L_e$ (km) | multiple | **0.45** | **0.85** | $L_e = 7.38(\pm6.13) + \mathbf{0.71}(\pm0.30) \cdot T + \mathbf{1.77}(\pm0.23) \cdot U_w$ (1) | $R_c^2$=0.71 RMSE=5.88 km RRMSE=19.2% |
| | simple | | **0.82** | $L_e = -2.18(\pm4.98) + \mathbf{1.51}(\pm0.22) \cdot U_w$ (2) | $R^2$=0.67 RMSE=6.57 km RRMSE=21.5% |
| $S_p$ (km²) | multiple | 0.37 | **0.75** | $S_p = -37.5(\pm343.0) + 32.1(\pm17.0) \cdot T + \mathbf{68.6}(\pm12.9) \cdot U_w$ (3) | $R_c^2$=0.53 RMSE=329 km² RRMSE=42.9% |
| | simple | | **0.71** | $S_p = -469.0(\pm269.0) + \mathbf{56.8}(\pm11.9) \cdot U_w$ (4) | $R^2$=0.50 RMSE=355 km² RRMSE=46.2% |
| $C$ (-) | multiple | **-0.44** | **0.76** | $C = \mathbf{0.206}(\pm0.054) - \mathbf{0.006}(\pm0.003) \cdot T + \mathbf{0.011}(\pm0.002) \cdot U_w$ (5) | $R_c^2$=0.66 RMSE=0.06 RRMSE=9.0 % |
| | simple | | **0.82** | $C = \mathbf{0.288}(\pm0.044) + \mathbf{0.013}(\pm0.002) \cdot U_w$ (6) | $R^2$=0.67 RMSE=0.06 RRMSE=10.0 % |
| $Md(X_{FS})$ (km) | multiple | **0.76** | -0.47 | $Md(X_{FS}) = \mathbf{2.52}(\pm0.19) + \mathbf{0.05}(\pm0.01) \cdot T - \mathbf{0.02}(\pm0.01) \cdot U_w$ (7) | $R_c^2$=0.73 RMSE=0.18 km RRMSE=18.0% |
| | simple | **0.79** | | $Md(X_{FS}) = \mathbf{2.18}(\pm0.18) + \mathbf{0.05}(\pm0.01) \cdot T$ (8) | $R^2$=0.62 RMSE=0.21 km RRMSE=21.7% |



**Table 4. Statistical characteristics of simple and multiple regression for estimating coefficients of distance transformation ($t_1$,$t_2$) (Eq. 10) on the basis of meteorological conditions: wind speed ($U_w$) and temperature ($T$) in 24h period preceding satellite overpass. For multiple regression partial correlation coefficients and corrected determination coefficients ($R_c^2$) are given, simple regression is determined for the selected independent variable with a higher correlation coefficient, ignoring the influence of the second one; coefficients statistically significant at $p<0.05$ are bolded; for regression coefficients standard errors of estimation are given in parentheses.**

| dependent variable | regression | correlation coefficients | | regression equation | regression statistics |
|---|---|---|---|---|---|
| | | $T$ (°C) | $U_w$ (m·s$^{-1}$) | | |
| $t_1$ | multiple | **-0.60** | 0.35 | $t_1 = -0.186(\pm0.102) - \mathbf{0.018}(\pm0.005) \cdot T + 0.007(\pm0.004) \cdot U_w$ (11) | $R_c^2$=0.52 RMSE=0.10 RRMSE=29.0% |
| | simple | **-0.69** | | $t_1 = -0.093(\pm0.089) - \mathbf{0.020}(\pm0.004) \cdot T$ (12) | $R^2$=0.47 RMSE=0.11 RRMSE=29.8% |
| $t_2$ | multiple | 0.33 | **-0.60** | $t_2 = \mathbf{1.128}(\pm0.120) + 0.010(\pm0.006) \cdot T - 0.016(\pm0.005) \cdot U_w$ (13) | $R_c^2$=0.51 RMSE=0.12 RRMSE=20.1% |
| | simple | | **-0.70** | $t_2 = \mathbf{0.995}(\pm0.093) - \mathbf{0.019}(\pm0.004) \cdot U_w$ (14) | $R^2$=0.49 RMSE=0.12 RRMSE=21.3% |