# Peer review of "Spatial characteristics of frazil streaks in the Terra Nova Bay Polynya from high-resolution visible satellite imagery"

_EGUsphere, 2022_

## Author Comment (AC1)

**RC1**: 'Comment on egusphere-2022-1435', Anonymous Referee #1, 01 Feb 2023

This is comprehensive study on the spatial characteristics of frazil streaks using high resolution satellite imagery and how the observed characteristics relate to temperature and wind speed. Some interesting new results described and generally well written, only a few minor changes are needed in the text to be ready for publication.

Response: We would like to thank for your review of the manuscript and providing your comments and suggestions to improve the quality of the manuscript. The following responses (in blue font) have been prepared to address all comments point by point.

**General comments**

You include some results at the end of introduction in the final paragraph, eg 'as expected ice concentration increases with X faster at lower T…' This is quite densely written, and I think just an overview of what you are going to do but not the relationships you find is what is needed at the point in the paper. You could move this to a summary after the discussion, as the discussion is relatively lengthy (which is fine) so a shorter summary of findings afterwards reiterating the main results to finish would help round off the paper nicely.

Response: As suggested, this part of the text will be moved to the end of the discussion to summarize it. Instead, at the end of the Introduction we will add a few sentences describing the purpose and scope of the work:

The aim of this study is to characterize geometric features of frazil streaks formed in polynyas, based on high-resolution (pixel size 10–15 m) visible satellite imagery recorded over the Terra Nova Bay (TNB; Fig. 1). Polynya size, ice concentration, and geometric properties of streaks are determined and related to the observed air temperature, wind speed and direction to explain observed patterns of frazil at the sea surface and to find simple empirical formulae linking the atmospheric forcing and analysed variables. Additionally, for a subset of satellite scenes in which wind waves are discernible, peak wave length and direction are determined and compared with corresponding open-water wave growth curves computed with a spectral wave model in order to analyse how wave interactions with frazil/grease ice may influence wave growth.

A bit more explanation/suggestion of what the mathematical relationships mean physically would help the paper flow and ensure the findings are clear throughout the results section and make it easier to read.

Response: The relationships found in our analysis are purely data-based, that is, they have statistical character. Their general form agrees with our qualitative understanding of the underlying processes – and we do provide that type of explanation in the 'results' and 'discussion' sections. For example, we do discuss the relationship between wind speed, air temperature, and changes of frazil concentration with offshore distance. However, as any theory relating, for example, the orientation of streaks to the combined wave-, wind and buoyancy-induced circulation in the mixed layer is lacking, we intentionally avoid any deeper physical interpretation – it would have a very speculative character. We believe that our data can help to develop such models and theories in the future.

And in the figure captions I think it would read easier if instead of putting 'what is in the subplot' (a), it was '(a) what is in the subplot….'

Response: All captions will be revised to make them more readable

And I had a few general questions about frazil streaks, though I appreciate that you focused on spatial characteristics, but I wondering whether is it known how thick/deep the frazil streaks are and what their vertical structure is like? Do you have any idea how important this might be or how it might interact with the horizontal features? And what is the timescale for the development of the streaks, how long do they stick around?

Response: The satellite data used here provide information only about the surface of the polynya without insight into its vertical structure. In addition, their temporal resolution does not allow inferences about changes over time, giving only single snapshots of streak development. Thus, the issues you mentioned are beyond the scope of this work. Moreover, very little is known about the 3D distribution of frazil in the mixed layer, and its interactions with mixed layer dynamics. To the best of our knowledge, the observational and modelling works we cite throughout the paper (and, in particular, in the introduction) are the only ones that address those questions. As we write in the text, models and observations consistently report the presence of frazil at depths of up to a few tens of meters, although its concentration decreases very fast with depth; see, e.g., Ohshima et al., (2022). However, none of the observational studies links streaks at the surface with the information at depth.
As far as the time scale of streaks' formation is concerned, there are no data from polynyas, but the Langmuir circulation itself (in ice-free regions) is known to develop and disappear very quickly in response to changes of the wind and wave forcing – within minutes or tens of minutes.

**Specific Comments**

Lines 112-127: At the end of the introduction you put a summary what you do in the study and some of the results. Instead an overview of what you are about to present and what is in each section would work better here. You could add a summary after your discussion/conclusion and move some of this text there. If possible and remaining consider writing the summary without symbol abbreviations.

Response above (the first general comment). We will revise the discussion to avoid abbreviations of symbols.

Fig 2: last figure has a different coloured background to the others which I don't think is intentional. Label the subplots a,b,c

Response: Thank you for pointing it out, it was not intentional, the figure will be corrected and subplots labelled.

Line 140: Sept and Oct – was this due to clouds? Or is this a time when they are most likely to be seen? Do we know much about their seasonal cycle?

Response: The opening of the TNBP is caused by katabatic winds, which are also a factor initiating the process of frazil formation. Satellite observations (e.g. Aulicino et al., 2018) indicate that alternating phases of opening and closing the polynya may occur very irregularly throughout the period of freezing, i.e. March-October. The lack of sunlight during the austral winter limits the use of optical sensors to the short beginning or end of that period. Hence the

available scenes are mainly for the months of September-October. Cloud cover further limits the number of scenes useful for analysis.

> Aulicino, G., Sansiviero, M., Paul, S., Cesarano, C., Fusco, G., Wadhams, P., and Budillon, G.: A new approach for monitoring the Terra Nova Bay Polynya through MODIS ice surface temperature imagery and its validation during 2010 and 2011 winter seasons, Remote Sens., 10, 366, doi:10.3390/rs10030366, 2018.

We will add an explanation in the text:

The lack of sunlight during the austral winter limits the use of optical sensors in frazil analysis mainly to the end of the freezing period. However, previous observations of the TNBP (e.g. Aulicino et al., 2018) indicate that alternating phases of opening and closing the polynya may occur very irregularly throughout the period of March-October, with no clear seasonal pattern.

Line 273: do you mean correlated with as opposed to 'consistent with the variability of the average wind speed'

Response: Yes, that's what we meant. For clarification, we'll replace the word "consistent" with "correlated."

Line 275: comment on 'the area of the polynya is less determined…' can you explain this statement?

Response: We meant a weaker correlation. We will correct the text:

However, the area of polynya is less **correlated with** meteorological conditions than its maximum extent.

Line 279: Potentially add a sentence clarifying that wind speed is suggested to be a dominant influence on extent and frazil concentration. You say it already, but wouldn't harm to state it clearly after the details.

Response: We will add the sentence as suggested

Line 311: I don't think that wind speed is given in either figure listed, I don't understand the comment. Sorry if I have missed something

Response: Indeed, the figures we refer to do not show the wind speed, but the RMSE values for specific situations. The comment results from the comparison of this Figure 5 or Suppl. Fig. S4 with Figure 2a. We will remove this statement and correct the reference to the Figures in the next sentence to clarify this:

Polynya events with average wind speed higher than 25 m·s$^{-1}$ have an RMSE below 0.06 (**compare Fig. 5 or Suppl. Fig. S4 with Fig. 2a**). Larger residuals occur when weather conditions are relatively mild and ice transformation can be observed within the polynya, or if consolidated ice made it difficult to delineate the polynya edge.

Line 319: 'the positive correlation between wind speed and the overall extent of the polynya' do you mean that increased wind speeds increases the polynya extent which decreases the frazil concentration (frazil is more spread out)? Could you explain a little here

Response: Yes, the wind speed controls the surface currents and, through waves, the Stokes drift, thus controlling the speed of ice drift

Line 327: Do you think that incorporating more observations will increase the percentage of variance explained? And can you comment on what might cause the rest, and the role of the oceanic/mixed layer conditions and potential timescales involved?

Response: Including more observations may increase the percentage of variance explained, but rather by a small amount. This is due to both the large simplification of the proposed relationship (taking into account only the wind speed and temperature as explanatory variables) and the accuracy of determining the frazil concentration during events where the boundaries of polynya are not distinct. What we are talking about in this section is how to better fit the regression line, which in Figure 6a appears to be slightly shifted towards the outermost values. To explain a higher percentage of variance, additional explanatory variables would have to be included to take into account for, for example, the development of the polynya over time or factors that locally modify ice drift.

Line 359: is 50 m the limit of the previous study? If so, state that

Response: The authors of earlier studies do not specify the width of the analysed streaks, they only describe the distances between them. However, taking into account that the SAR data they use has a resolution of 100 m, it can be assumed that bands with a width of less than half a pixel width, even strongly contrasting with water, will not be visible in these images, hence the limit of 50 m. The choice of such a limit is also justified by the fact that such narrow bands in our research dominated the zone up to 3-4 km from the shore. In the cited paper of Ciappa and Pietranera (2013), the authors did not observe any frazil streaks in the SAR images in this zone.

Line 412: 'forcing', maybe phrase this as atmospheric conditions or some variations on atmospheric/sea ice/OML conditions

Response: 'forcing' will be clarified:

The available satellite and other data show that the shape and extent of polynyas evolve on a daily or even hourly basis (Kern et al. 2007, Ciappa and Pietranera, 2013, Aulicino et al., 2018), indicating that processes taking place there are nonstationary and very sensitive to changes in the **forcing, i.e. atmospheric conditions and sea ice drift.**

Line 425: I'm not sure what you mean by 'high dynamics'

Response: The cited papers, which analysed the fluctuations of the area of polynya in TNB, show that its opening is most often a short-term event, lasting from several hours to several days. Subsequent openings are separated by similarly short closing phases. By 'high dynamics' we meant frequent changes in the extent of the polynya. The sentence will be rewritten to clarify this:

Considering **highly dynamic, intermittent character of polynya formation and disappearance** described in previous studies (Kern et al. 2007, Ciappa and Pietranera, 2012, Aulicino et al., 2018), it is expected that alternating phases of those processes often lead to such an uneven **frazil accumulation along the downwind edge of the polynya.**

Line 458: Redefine Cz(X), and any other symbols you used in the discussion
Line 482: redefine the angles you are comparing

Response to both: We will revise the discussion to avoid symbols and abbreviations. If necessary, all symbols appearing in the discussion will be explained again. Additionally, a list of symbols with their definitions will be added in the Appendix.

Thank you for the other style suggestions and corrections. All will be revised and corrected accordingly:

Line 125: remove extra space before 'The dominant role…'
Line 294: Change 'also indicates an area' to 'is in an area'
Line 312: Do you mean 'residuals' instead of 'residues'
Line 325: delete 'It seems that taking into account'
Line 331: delete extra space after '160 m'
Line 353: I think you mean 'extend' instead of 'extent'
Line 415: redefine OML as ocean mixed layer here
Line 428: Remove the long bracket and split into a sentence so it reads more smoothly. Could put the relevant features in a sentence afterwards with something like 'The features that demonstrate are' or similar
Line 443: Change 'stronger' to 'more strongly', it isn't currently wrong but I think it will flow better
Line 478: delete 'As said'
Line 497: Remove bracket around sentence
Line 517: Remove bracket from around sentence.

---

## Author Comment (AC2)

**RC2**: 'Comment on egusphere-2022-1435', Anonymous Referee #2, 10 Feb 2023

The paper contains a very thorough analysis of frazil ice streaks in a large polynya using the possibilities of remote sensing. It is a good source of statistical information for further investigation. I particularly like the inclusion of wave modelling, which allows for understanding the significance of the findings better. I recommend publication with minor revisions.

Response: We would like to thank for your review of the manuscript and providing your comments and suggestions to improve the quality of the manuscript. The following responses (in blue font) have been prepared to address all comments point by point.

There are a lot of symbols used in the text, a table with the variable definitions would be helpful.

Response: A table with a list of symbols used and their definitions will be added as an Appendix

The paper also ends quite abruptly, one or two concluding sentences could be nice.

Response below along with the answer to the comment on line 112

**Specific comments:**

Line 30: perhaps an estimate of what percentage of the heat and moisture fluxes in the ice cover of this region happen over the polynya

Response: We will add relevant literature references:

Due to very strong ocean–atmosphere heat and moisture fluxes and high rates of new ice production with the associated brine rejection, latent heat polynyas play an important role in shaping the local and regional weather patterns, as well as in water mass formation, ocean mixing and baroclinic processes **(Nakata et al., 2015, Ohshima et al., 2022)**.

Line 112 onward: the introduction has changed into a summary, please put this at the end of the paper

Response: As suggested, this part of the text will be moved to the end of the discussion to summarize it. Instead, at the end of the Introduction we will add a few sentences describing the purpose and scope of the work:

The aim of this study is to characterize geometric features of frazil streaks formed in polynyas, based on high-resolution (pixel size 10–15 m) visible satellite imagery recorded over the Terra Nova Bay (TNB; Fig. 1). Polynya size, ice concentration, and geometric properties of streaks are determined and related to the observed air temperature, wind speed and direction to explain observed patterns of frazil at the sea surface and to find simple empirical formulae linking the atmospheric forcing and analysed variables. Additionally, for a subset of satellite scenes in which wind waves are discernible, peak wave length and direction are determined and compared with corresponding open-water wave growth curves computed with a spectral wave model in order to analyse how wave interactions with frazil/grease ice may influence wave growth.

Line 145: the correlation with meteorological data on particular timescales is a result in itself, so this sentence belongs in the results section

Response: We agree, but have chosen not to consider correlations with meteorological data on different time scales. This statement has been included in paragraph "2.1 Dataset" only to explain the length of the meteorological data averaging period. Results of correlations for this particular timescale are presented in the results section.

Line 225: do you mean "For three events"? These specific events have not been introduced before I believe

Response: Thank you for pointing it out. This will be corrected and supplemented with the dates of these events:

**For three events (6 and 22 Oct 2016, 8 Oct 2019)**, there are coinciding pairs of MSI/OLI images, recorded at a time interval of no more than 0.5 hours.

Thank you for the other style suggestions and corrections. All will be revised and corrected accordingly:

Line 9: "as well as" humidity instead of "and"
Line 26: use commas when using "or" for synonyms: Coastal, or latent heat, polynyas etc.
Line 35: please split this sentence in two to make it easier to read
Line 312: "with average"
Line 328: "is shown"
Line 331: "changes"

---

## Author Response (AR1)

**RC1**: 'Comment on egusphere-2022-1435', Anonymous Referee #1, 01 Feb 2023

This is comprehensive study on the spatial characteristics of frazil streaks using high resolution satellite imagery and how the observed characteristics relate to temperature and wind speed. Some interesting new results described and generally well written, only a few minor changes are needed in the text to be ready for publication.

Response: We would like to thank for your review of the manuscript and providing your comments and suggestions to improve the quality of the manuscript. The following responses (in blue font) have been prepared to address all comments point by point.

**General comments**

You include some results at the end of introduction in the final paragraph, eg 'as expected ice concentration increases with X faster at lower T…' This is quite densely written, and I think just an overview of what you are going to do but not the relationships you find is what is needed at the point in the paper. You could move this to a summary after the discussion, as the discussion is relatively lengthy (which is fine) so a shorter summary of findings afterwards reiterating the main results to finish would help round off the paper nicely.

Response: As suggested, this part of the text has been moved to the 'Discussion' to highlight what was done (in the revised manuscript lines: 417-419, 487-488) and to summarize discussion (lines: 553-566). Instead, at the end of the Introduction we added a few sentences describing the purpose and scope of the work (lines: 113-120):

The aim of this study is to characterize geometric features of frazil streaks formed in polynyas, based on high-resolution (pixel size 10–15 m) visible satellite imagery recorded over the Terra Nova Bay (TNB; Fig. 1). Polynya size, ice concentration, and geometric properties of streaks are determined and related to the observed air temperature, wind speed and direction to explain observed patterns of frazil at the sea surface and to find simple empirical formulae linking the atmospheric forcing and analysed variables. Additionally, for a subset of satellite scenes in which wind waves are discernible, peak wave length and direction are determined and compared with corresponding open-water wave growth curves computed with a spectral wave model in order to analyse how wave interactions with frazil/grease ice may influence wave growth.

A bit more explanation/suggestion of what the mathematical relationships mean physically would help the paper flow and ensure the findings are clear throughout the results section and make it easier to read.

Response: The relationships found in our analysis are purely data-based, that is, they have statistical character. Their general form agrees with our qualitative understanding of the underlying processes – and we do provide that type of explanation in the 'results' and 'discussion' sections. For example, we do discuss the relationship between wind speed, air temperature, and changes of frazil concentration with offshore distance (lines: 464-474). However, as any theory relating, for example, the orientation of streaks to the combined wave-, wind and buoyancy-induced circulation in the mixed layer is lacking, we intentionally avoid any deeper physical interpretation – it would have a very speculative character. We believe that our data can help to develop such models and theories in the future.

And in the figure captions I think it would read easier if instead of putting 'what is in the subplot' (a), it was '(a) what is in the subplot….'

Response: All captions has been revised as suggested to make them more readable

And I had a few general questions about frazil streaks, though I appreciate that you focused on spatial characteristics, but I wondering whether is it known how thick/deep the frazil streaks are and what their vertical structure is like? Do you have any idea how important this might be or how it might interact with the horizontal features? And what is the timescale for the development of the streaks, how long do they stick around?

Response: The satellite data used here provide information only about the surface of the polynya without insight into its vertical structure. In addition, their temporal resolution does not allow inferences about changes over time, giving only single snapshots of streak development. Thus, the issues you mentioned are beyond the scope of this work. Moreover, very little is known about the 3D distribution of frazil in the mixed layer, and its interactions with mixed layer dynamics. To the best of our knowledge, the observational and modelling works we cite throughout the paper (in particular, in the introduction) are the only ones that address those questions. As we write in the text, models and observations consistently report the presence of frazil at depths of up to a few tens of meters, although its concentration decreases very fast with depth; see, e.g., Ohshima et al., (2022). However, none of the observational studies links streaks at the surface with the information at depth.
As far as the time scale of streaks' formation is concerned, there are no data from polynyas, but the Langmuir circulation itself (in ice-free regions) is known to develop and disappear very quickly in response to changes of the wind and wave forcing – within minutes or tens of minutes.

**Specific Comments**

Lines 112-127: At the end of the introduction you put a summary what you do in the study and some of the results. Instead an overview of what you are about to present and what is in each section would work better here. You could add a summary after your discussion/conclusion and move some of this text there. If possible and remaining consider writing the summary without symbol abbreviations.

Response above (the first general comment). We revised the discussion to avoid abbreviations and symbols.

Fig 2: last figure has a different coloured background to the others which I don't think is intentional. Label the subplots a,b,c

Response: Thank you for pointing it out, it was not intentional, the figure has been corrected, and subplots labelled.

We also found that Figure 9e,f presented $C_s$ (the same results as in Fig. 3c,d) instead of $\theta_{FS}$. It has been corrected. Other figures have been adjusted (font, symbols, resolution, etc.) to the requirements to make them more readable.

Line 140: Sept and Oct – was this due to clouds? Or is this a time when they are most likely to be seen? Do we know much about their seasonal cycle?

Response: The opening of the TNBP is caused by katabatic winds, which are also a factor initiating the process of frazil formation. Satellite observations (e.g. Aulicino et al., 2018) indicate that alternating phases of opening and closing the polynya may occur very irregularly throughout the period of freezing, i.e. March-October. The lack of sunlight during the austral winter limits the use of optical sensors to the short beginning or end of that period. Hence the available scenes are mainly for the months of September-October. Cloud cover further limits the number of scenes useful for analysis.

We added an explanation in the text (lines: 135-138):

The lack of sunlight during the austral winter limits the use of optical sensors in frazil analysis mainly to the end of the freezing period. However, previous observations of the TNBP (e.g. Aulicino et al., 2018) indicate that alternating phases of opening and closing the polynya may occur very irregularly throughout the period of March-October, with no clear seasonal pattern.

Line 273: do you mean correlated with as opposed to 'consistent with the variability of the average wind speed'

Response: Yes, that's what we meant. For clarification, we replaced the word "consistent" with "correlated" (line 273)

Line 275: comment on 'the area of the polynya is less determined…' can you explain this statement?

Response: We meant a weaker correlation. We corrected the text (line 274):

However, the area of polynya is less **correlated with** meteorological conditions than its maximum extent.

Line 279: Potentially add a sentence clarifying that wind speed is suggested to be a dominant influence on extent and frazil concentration. You say it already, but wouldn't harm to state it clearly after the details.

Response: We added the sentence as suggested (lines: 279-280)

Line 311: I don't think that wind speed is given in either figure listed, I don't understand the comment. Sorry if I have missed something

Response: Indeed, the figures we refer to do not show the wind speed, but the RMSE values for specific situations. The comment results from the comparison of this Figure 5 or Suppl. Fig. S4 with Figure 2a. We removed this statement and corrected the reference to the Figures in the next sentence to clarify this (line 313):

Polynya events with average wind speed higher than 25 m·s$^{-1}$ have an RMSE below 0.06 (**compare Fig. 5 or Suppl. Fig. S4 with Fig. 2a**). Larger residuals occur when weather conditions are relatively mild and ice transformation can be observed within the polynya, or if consolidated ice made it difficult to delineate the polynya edge.

Line 319: 'the positive correlation between wind speed and the overall extent of the polynya' do you mean that increased wind speeds increases the polynya extent which decreases the frazil concentration (frazil is more spread out)? Could you explain a little here

Response: Yes, the wind speed controls the surface currents and, through waves, the Stokes drift, thus controlling the speed of ice drift. We added an explanation (lines: 320-321):

The latter effect is related to the positive correlation between wind speed and the overall extent of the polynya, **which is the result of the influence of the wind on the speed of ice drift**.

Line 327: Do you think that incorporating more observations will increase the percentage of variance explained? And can you comment on what might cause the rest, and the role of the oceanic/mixed layer conditions and potential timescales involved?

Response: Including more observations may increase the percentage of variance explained, but rather by a small amount. This is due to both the large simplification of the proposed relationship (taking into account only the wind speed and temperature as explanatory variables) and the accuracy of determining the frazil concentration during events where the boundaries of polynya are not distinct. What we are talking about in this section is how to better fit the regression line, which in Figure 6a appears to be slightly shifted towards the outermost values. To explain a higher percentage of variance, additional explanatory variables would have to be included to take into account for, for example, the development of the polynya over time or factors that locally modify ice drift.

Line 359: is 50 m the limit of the previous study? If so, state that

Response: The authors of earlier studies do not specify the width of the analysed streaks; they only describe the distances between them. However, taking into account that the SAR data they use has a resolution of 100 m, it can be assumed that bands with a width of less than half a pixel width, even strongly contrasting with water, will not be visible in these images, hence the limit of 50 m. The choice of such a limit is also justified by the fact that such narrow bands in our research dominated the zone up to 3-4 km from the shore. In the cited paper of Ciappa and Pietranera (2013), the authors did not observe any frazil streaks in the SAR images in this zone.

Line 412: 'forcing', maybe phrase this as atmospheric conditions or some variations on atmospheric/sea ice/OML conditions

Response: 'forcing' has been clarified (line 413):

The available satellite and other data show that the shape and extent of polynyas evolve on a daily or even hourly basis (Kern et al. 2007, Ciappa and Pietranera, 2013, Aulicino et al., 2018), indicating that processes taking place there are nonstationary and very sensitive to changes in the **forcing, i.e. atmospheric conditions and sea ice drift.**

Line 425: I'm not sure what you mean by 'high dynamics'

Response: The cited papers, which analysed the fluctuations of the area of polynya in TNB, show that its opening is most often a short-term event, lasting from several hours to several days. Subsequent openings are separated by similarly short closing phases. By 'high dynamics'

we meant frequent changes in the extent of the polynya. The sentence has been rewritten to clarify this (lines: 429-432):

Considering **highly dynamic, intermittent character of polynya formation and disappearance** described in previous studies (Kern et al. 2007, Ciappa and Pietranera, 2012, Aulicino et al., 2018), it is expected that alternating phases of those processes often lead to such an uneven **frazil accumulation along the downwind edge of the polynya.**

Line 458: Redefine Cz(X), and any other symbols you used in the discussion
Line 482: redefine the angles you are comparing

Response to both: We revised the discussion to avoid symbols and abbreviations. If necessary, all symbols appearing in the discussion are explained again. Additionally, a list of symbols with their definitions has been added in the Appendix.

Thank you for the other style suggestions and corrections. All has been corrected accordingly (the corresponding lines of the revised manuscript are given in parentheses):

Line 125: remove extra space before 'The dominant role…' (moved to line 560)
Line 294: Change 'also indicates an area' to 'is in an area' (line 295)
Line 312: Do you mean 'residuals' instead of 'residues' (line 313)
Line 325: delete 'It seems that taking into account' (line 326)
Line 331: delete extra space after '160 m' (line 332)
Line 353: I think you mean 'extend' instead of 'extent' (line 295)
Line 415: redefine OML as ocean mixed layer here (line 416)
Line 428: Remove the long bracket and split into a sentence so it reads more smoothly. Could put the relevant features in a sentence afterwards with something like 'The features that demonstrate are' or similar (lines: 435-438)
Line 443: Change 'stronger' to 'more strongly', it isn't currently wrong but I think it will flow better (line 449)
Line 478: delete 'As said' (line 485)
Line 497: Remove bracket around sentence (lines: 506-507)
Line 517: Remove bracket from around sentence (lines: 526-529).

We also checked and corrected our manuscript for typos and other errors.

**RC2**: 'Comment on egusphere-2022-1435', Anonymous Referee #2, 10 Feb 2023

The paper contains a very thorough analysis of frazil ice streaks in a large polynya using the possibilities of remote sensing. It is a good source of statistical information for further investigation. I particularly like the inclusion of wave modelling, which allows for understanding the significance of the findings better. I recommend publication with minor revisions.

Response: We would like to thank for your review of the manuscript and providing your comments and suggestions to improve the quality of the manuscript. The following responses (in blue font) have been prepared to address all comments point by point.

There are a lot of symbols used in the text, a table with the variable definitions would be helpful.

Response: A table with a list of symbols used and their definitions has been added as an Appendix

The paper also ends quite abruptly, one or two concluding sentences could be nice.

Response below along with the answer to the comment on line 112

**Specific comments:**

Line 30: perhaps an estimate of what percentage of the heat and moisture fluxes in the ice cover of this region happen over the polynya

Response: We added relevant literature references (in the revised manuscript line 32):

Due to very strong ocean–atmosphere heat and moisture fluxes and high rates of new ice production with the associated brine rejection, latent heat polynyas play an important role in shaping the local and regional weather patterns, as well as in water mass formation, ocean mixing and baroclinic processes **(Nakata et al., 2015, Ohshima et al., 2022)**.

Line 112 onward: the introduction has changed into a summary, please put this at the end of the paper

As suggested, this part of the text has been moved to the 'Discussion' to highlight what was done (in the revised manuscript lines: 417-419, 487-488) and to summarize discussion (lines: 553-566). Instead, at the end of the Introduction we added a few sentences describing the purpose and scope of the work (lines: 113-120):

The aim of this study is to characterize geometric features of frazil streaks formed in polynyas, based on high-resolution (pixel size 10–15 m) visible satellite imagery recorded over the Terra Nova Bay (TNB; Fig. 1). Polynya size, ice concentration, and geometric properties of streaks are determined and related to the observed air temperature, wind speed and direction to explain observed patterns of frazil at the sea surface and to find simple empirical formulae linking the atmospheric forcing and analysed variables. Additionally, for a subset of satellite scenes in which wind waves are discernible, peak wave length and direction are determined and compared with corresponding open-water wave

growth curves computed with a spectral wave model in order to analyse how wave interactions with frazil/grease ice may influence wave growth.

Line 145: the correlation with meteorological data on particular timescales is a result in itself, so this sentence belongs in the results section

Response: We agree but have chosen not to consider correlations with meteorological data on different time scales. This statement has been included in paragraph "2.1 Dataset" only to clarify the length of the meteorological data averaging period that we used. Results of correlations for this particular timescale are presented in the 'Results' section.

Line 225: do you mean "For three events"?  These specific events have not been introduced before I believe

Response: Thank you for pointing it out. This has been corrected and supplemented with the dates of these events (line: 223-224):

**For three events (6 and 22 Oct 2016, 8 Oct 2019)**, there are coinciding pairs of MSI/OLI images, recorded at a time interval of no more than 0.5 hours.

Line 331: "changes"

We refer to ranges of two statistics: median and IQR, thus we corrected as follows (lines: 332-334):

In the analysed cases, the median **values** and the interquartile range **values** of the streak widths change in the **ranges** of 40–60 m and 60–160 m, respectively.

Thank you for the other style suggestions and corrections. All has been corrected accordingly (the corresponding lines of the revised manuscript, if changed, are given in parentheses):

Line 9: "as well as" humidity instead of "and"
Line 26: use commas when using "or" for synonyms: Coastal, or latent heat, polynyas etc.
Line 35: please split this sentence in two to make it easier to read (lines: 36-39)
Line 312: "with average"
Line 328: "is shown" (line 329)

We also found that Figure 9e,f presented $C_s$ (the same results as in Fig. 3c,d) instead of $\theta_{FS}$. It has been corrected. Other figures have been adjusted (font, symbols, resolution, etc.) to the requirements to make them more readable. We also checked and corrected our manuscript for typos and other errors.